# Frontier LLMs Still Struggle with Simple Reasoning Tasks

## Abstract

While state-of-the-art large language models (LLMs) demonstrate advanced reasoning capabilities—achieving remarkable performance on challenging competitive math and coding benchmarks—they also frequently fail on tasks that are easy for humans. This work studies the performance of frontier LLMs on a broad set of such "easy" reasoning problems. By extending previous work in the literature, we create a suite of *procedurally generated* simple reasoning tasks, including counting, first-order logic, proof trees, and travel planning, with changeable parameters (such as document length. or the number of variables in a math problem) that can arbitrarily increase the amount of computation required to produce the answer while preserving the fundamental difficulty. While previous work showed that traditional, non-thinking models can be made to fail on such problems, we demonstrate that even state-of-the-art thinking models consistently fail on such problems and for similar reasons (e.g., statistical shortcuts, errors in intermediate steps, and difficulties in processing long contexts). To further understand the behavior of the models, we introduce the Unpuzzles dataset, a different "easy" benchmark consisting of trivialized versions of well-known math and logic puzzles. Interestingly, while modern LLMs excel at solving the original puzzles, they tend to fail on the trivialized versions, exhibiting several typical failure patterns related to memorizing the originals. We show that this happens even if the models are otherwise able to solve problems with different descriptions but requiring the same logic. Our results highlight that out-of-distribution generalization is still problematic for frontier language models and the new generation of thinking models, even for simple reasoning tasks, and making tasks easier does not necessarily imply improved performance.

## 1 Introduction

Modern transformer-based large language models (LLMs) (Vaswani, 2017) trained using next-token prediction have achieved significant success across a wide range of tasks, especially in reasoning. For instance, OpenAI's o1 model—one of the leading reasoning models to date—"ranks in the 89th percentile on competitive programming questions (Codeforces), places among the top 500 students in the US in a qualifier for the USA Math Olympiad (AIME), and exceeds human PhD-level accuracy on a benchmark of physics, biology, and chemistry problems (GPQA)".[1]

On the other hand, researchers continue to uncover surprisingly simple reasoning problems that still confuse even the most advanced LLMs. These include tasks such as counting characters in words, comparing numbers like 9.11 and 9.9 (Xie, 2024), making simple inferences about family relationships (Nezhurina et al., 2024), and solving various classes of arithmetic and logic problems (see, e.g., McLeish et al., 2024; Zhang et al., 2022). Many of these failures are identified in isolation, making it difficult to find common underlying issues. Moreover, some studies focus on earlier model generations, leaving it open whether these failures persist in state-of-the-art (SOTA) models.

In this work, we study the performance of several high-quality, open and closed-source language models, both traditional (GPT-4o, Gemini 1.5 Pro, 2.0 Flash, and 2.5 Flash, Gemma 3 27B, Claude 3.5 and 3.7 Sonnet) and thinking variants (OpenAI o1 and o3, Gemini 2.0 Flash Thinking, 2.5 Pro and

---

[1]https://openai.com/index/learning-to-reason-with-llms/

---

**Original Puzzle**

13 purple, 15 yellow, and 17 maroon chameleons are found on an island. When two different-coloured chameleons meet in a pair, they both turn into the third color. Is it possible that, after some pairwise meetings, all the chameleons are the same color?
**Answer**: no (proof by contradiction related to a problem invariant)

**Unpuzzle**

15 purple, 15 yellow, and 17 maroon chameleons are found on an island. When two different-coloured chameleons meet in a pair, they both turn into the third color. Is it possible that, after some pairwise meetings, all the chameleons are the same color??
**Answer**: yes (purple and yellow chameleons all pair up)

**Context-shifted Unpuzzle**

There are 31 Spurs fans, 31 Arsenal fans, and 49 Chelsea fans. Every time fans of two different sports teams meet, they realize they are both wrong and become fans of the third team. Is it possible that, after a certain number of pairwise meetings, everyone is a fan of one team?
**Answer**: yes (Spurs fans and Arsenal fans all pair up)

Figure 1: Chameleons go on a date: a puzzle, corresponding unpuzzle, and a context-shifted unpuzzle

3.0 Pro, DeepSeek R1), across a broad range of "easy" reasoning problems. We begin by examining four simple reasoning tasks: (1) character and word counting, (2) first-order logic evaluation and negation, (3) math word problems based on proof trees, and (4) travel planning problems. Rather than using fixed datasets, we generate problems randomly and procedurally, incorporating tunable parameters—such as paragraph length in word counting and the number of cities in travel planning—that adjust the amount of computation required to produce an answer while preserving the underlying reasoning difficulty. With appropriately chosen parameters, these tasks may be tedious for humans but remain straightforward. On the other hand, frontier LLMs consistently fail on such tasks, with underlying causes including statistical shortcuts, errors in intermediate steps, and difficulties in dealing with long contexts. While previous work showed that earlier SOTA models fail on similar tasks, here we demonstrate that even the next generation of LLMs, the so-called *thinking models* fail when the tasks become long enough. To our knowledge, no earlier papers demonstrated that thinking models are similarly subject to such performance degradation; essentially, we provide evidence that many of the claims in the literature of decreasing LLM performance with task difficulty will apply to thinking models as well. Concurrently with our work, Shojaee et al. (2025) evaluate LLMs on four puzzles with programmable complexity, and show that thinking LLMs completely fail beyond a certain critical complexity threshold. However, the experiment design has been found lacking (Lawsen, 2025; Chan, 2025).

Analyzing the behavior of the models, we found that the main reasons of the failures can be attributed to procedure errors (i.e., making a mistake in applying a reasoning step), omitting information given in the task, incorrectly copying information in the reasoning trace, parsing problems (e.g., parsing parentheses in logic formulas), hallucinations, applying shortcuts and heuristics which do not generalize to the test data, and giving up the problem before a solution is reached. Tokenization can also be a source of problems, especially for character counting. Interestingly, we find that thinking models apply less shortcuts and are less prone to omissions, which seem to be the major sources of their superior behavior.

We can also found indications that some of the erroneous behavior stems from overfitting or, in other words, relying too much on memorization (which is also a natural source of failures of any machine learning model. To examine this issue better, we introduce a new dataset for evaluating language models called UNPUZZLES. This is a small dataset in two parts: The first has 97 well-known logical puzzles and brainteasers that are commonly found on the internet (and can be assumed to be in the training set of SOTA models), as well as their *trivialized* versions which we refer to as *"unpuzzles"*. Each unpuzzle is created manually by making minimal textual edits to the original puzzle in order to remove the difficulty and render the answer obvious. We demonstrate that while SOTA models all perform well on the original (difficult) puzzles, they exhibit poor performance on the corresponding (easy) unpuzzles. The second part focuses on a subset of 64 unpuzzles (with numerical answers that can be machine-evaluated) and adds a "context-shifted" version of the unpuzzle where the language,

setting, or vocabulary is changed but the logical structure is preserved. These context-shifted trivial problems can be used to test if a model has the ability of solving the simpler problems, which in turn helps us examine the reasons behind failures to solve the unpuzzle problems. An example is provided in Figure 1; the unpuzzle only differs by two characters, and the solution logic of the unpuzzle and the context-shifted version are identical.[2]

While many existing works evaluate LLM reasoning robustness by perturbing problems while maintaining the same difficulty level (Mirzadeh et al., 2024; McCoy et al., 2024), our study instead shows that *decreasing* difficulty can also lead to much worse performance. A key failure mode we observe is that LLMs tend to "overthink" easy problems, often erroneously reusing reasoning steps corresponding to the more complex puzzle solutions — a phenomenon we term *reasoning delirium*. Further, these failures are not because the models do not know how to reason about easy problems: every model we tested performed better on the context-shifted unpuzzles than the original ones, indicating that failure was at least in part due to memorization of the original puzzle.

In summary, we make the following contributions: (1) we conduct a comprehensive evaluation of frontier LLMs across a wide range of simple reasoning problems; (2) we connect failure modes to their potential causes; (3) we present a new set of procedurally generated reasoning tasks with tunable parameters that are challenging for high-quality LLMs; (4) we introduce the UNPUZZLES dataset that confuses frontier LLMs, exposing memorization artifacts. Our work demonstrates that the qualitative trend of performance degradation still exists even for the latest thinking models, even though quantitative results have improved. We hope the new benchmarks and our methodology for identifying failures will improve the assessment of reasoning capabilities of future model generations.

## 2 RELATED WORK

There is a long line of research focused on identifying tasks that challenge modern LLMs and developing new benchmarks. In this paper, we review the studies most relevant to the tasks we investigate. Transformer-based LLMs are known to struggle with seemingly simple tasks such as counting (Ouellette et al., 2023; Yehudai et al., 2024; Barbero et al., 2024) and copying (Liu et al., 2024; Barbero et al., 2024), due to issues related to tokenization, architecture, and embeddings. They also perform poorly on tasks requiring multi-step reasoning, such as arithmetic, logic puzzles, and dynamic programming (Dziri et al., 2024). The difficulty of solving simple logic problems has been explored in Yang et al. (2023); Parmar et al. (2024); Han et al. (2022), where these tasks are often framed as translation problems from natural language to first-order logic.

Other works, such as Valmeekam et al. (2024a;b), construct planning benchmarks using Planning Domain Definition Language (PDDL), while Xie et al. (2024) develops a travel planning benchmark in real-world scenarios. These studies show that existing LLMs are far from saturating these datasets. Additionally, reasoning benchmarks with large amounts of irrelevant content have been proposed (Shi et al., 2023; Mirzadeh et al., 2024) to test models' long-context generalization capabilities. Most of these benchmarks are fixed and often combine the core challenge (e.g., logic or planning) with secondary challenges, such as understanding PDDL or real-world commonsense reasoning for travel, making it difficult to pinpoint the exact sources of failure. Furthermore, fixed benchmarks are difficult to extend or generalize and are prone to saturation or overfitting as LLMs improve.

In contrast, our work takes a principled approach by simplifying problems to isolate failure causes. Our tasks are randomly and procedurally generated, allowing for easy adjustments to their distribution and difficulty (at a superficial level), ensuring they remain challenging for future LLMs. The work most relevant to ours is that of Opedal et al. (2025), which evaluates the out-of-distribution (OOD) generalization ability of LLMs through MathGAP, a framework that procedurally generates arithmetic problems by representing them as sequences of logical forms, with solutions structured as proof trees. Compared to Opedal et al. (2025), our work takes a broader perspective by examining a wider range of tasks and identifying multiple critical failure modes for OOD generalization. In the vision-languagemodel domain, Rahmanzadehgervi et al. (2025) consider problems where the models need to perform similarly simple problems on images, and identifies that image-processing mistakes cause degradation in visual/spatial reasoning.

---

[2]All data will be released.

In a concurrent work to ours, Shojaee et al. (2025) evaluate LLMs on four puzzles with controllable "complexity" and show that the accuracy of all models completely collapses beyond a certain complexity threshold. It has been pointed out that their experiment design is somewhat flawed, including unsolvable problems and potentially ignoring token limits (Lawsen, 2025; Chan, 2025).[3]. Nonetheless, we observe qualitatively similar performance with accuracy decreasing as a function of the task tediousness.

The idea of perturbing existing benchmarks to test the robustness of LLM reasoning has been explored in several prior works. Mirzadeh et al. (2024) introduce a variant of the GSM8K benchmark for mathematical reasoning, modifying numerical values and adding irrelevant information, both of which lead to a performance drop in common models. Similarly, Jiang et al. (2024) evaluate LLMs on conjunction and syllogistic fallacies by perturbing well-known problems—changing names, inserting celebrity references, adding irrelevant content, and replacing quantifiers with synonyms—revealing evidence of "token bias" in LLMs. The negative effect of adding irrelevant context to math word problems was also shown recently by Xu et al. (2025). These studies primarily focus on perturbing original problems while maintaining or increasing their difficulty.

In contrast, our UNPUZZLES benchmark takes the opposite approach: we make minimal edits to the wording but drastically *reduce* problem difficulty. A related work by Williams and Huckle (2024) introduces a benchmark of 30 easy problems that LLMs fail on, 12 of which are logical puzzles. Our evaluation on puzzles is considerably more comprehensive. Finally, the failure modes identified in UNPUZZLES also relate to findings from McCoy et al. (2024), which demonstrate that LLM accuracy is heavily influenced by the likelihood of task formulations, inputs, or outputs appearing in the training data.

## 3 PROCEDURALLY GENERATED REASONING TASKS

This section presents our collection of simple reasoning tasks, including several extensions of tasks from existing literature. Each task is procedurally generated, allowing a near-infinite number of new problems to be generated, and defined by parameters that control the difficulty or complexity. One of our goals was to design tasks that are straightforward (albeit tedious) for humans, but become unsolvable by frontier models when the difficulty parameters are large enough; all our results demonstrate this feature. For brevity, each task is described informally; full descriptions, usually with pseudocode, are in the appendix.

Throughout, we abbreviate Google's Gemini 1.5 Pro, 2.0 Flash, 2.0 Flash Thinking, 2.5 Pro, and 3.0 Pro with G1.5, G2.0F, G2.0FT, and G2.5P, G3, respectively. We also abbreviate Anthropic's Claude 3.5 and 3.7-sonnet (run without thinking tokens), OpenAI's o1, o3, GPT-4o, and GPT-5.1, DeepSeek's R1, and Gemma 3 27B by C3.5, C3.7, o1, o3, 4o, o5.1, R1, and gem3, respectively; see the appendix for the specific versions. Unless specified otherwise, for every task and every choice of parameters, we average the performance of the models across 20 randomly sampled tasks.

### 3.1 TASKS

**Character and word counting**  Until somewhat recently, many LLMs infamously could not count the number of r's in "strawberry." This task extends this task to simultaneous word or character counting. The WORD COUNTING task requires the model to simultaneously count the number of occurrences of each word in a list of size $k$ from a paragraph of length $m$. The task obviously becomes more difficult as $k$ and $m$ increase. The CHARACTER COUNTING task only requires counting a single character, which already proves difficult for the models. The paragraphs are extracted from the WikiText-2 dataset and are either selected to have minimum size $m = 50$ (with maximum size 150) or minimum size $m = 150$ (with maximum size 400).

**First-order logic tasks**  We evaluate models on two fundamental logic tasks: evaluating propositional logical statements and negating first-order logical statements. A logic formula can be represented with a tree with logic operators as nodes and propositions and predicates as leaves. An atomic proposition is a simple, binary-valued variable, usually represented $P$ or $Q$, whereas a

---

[3]While the work of Lawsen (2025) was initially published as a joke, some of the flaws discussed are legitimate concerns

predicate represents a property about an individual: for example, $P(x)$ indicates that individual $x$ has property $P$. We include the standard logical operators $\vee, \wedge, \Leftrightarrow, \Rightarrow, \neg, \forall x \in X$, and $\exists x \in X$, (respectively, or, and, equivalent, implies, negation, for all, and exists), where the last two are quantifying operators that require a domain to be specified. Exploiting the tree structure, we can sample a logic formula recursively. The complexity is controlled by the maximum depth $d$ and the total number $n$ of predicates and atomic propositions to sample for leaves. We either choose $n = 16$ (16 predicates, 16 atomic propositions, and 8 domains) or $n = 8$ (8, 8, and 4, respectively). The final parameter is what vocabulary we use for the leaves: we created three categories: random 20 character strings, capital letters (reflecting the training data), and words that describe motion pictures. We consider the tasks of (1) LOGIC EVALUATION - identifying which of four value assignments evaluates to true, and (2) LOGIC NEGATION - identifying the negation of a logic formula from four options. See Appendix H.2 and H.3 for more details and examples.

**Math word problems based on proof trees** We extend the MATHGAP task of Opedal et al. (2025), which uses a tree-based representation of proofs to generate mathematical word problems. Each problem is represented as a sequence of *logical forms* under the formalism from Opedal et al. (2024). A logical form is a truth statement about the world, typically describing an arithmetic relationship, such as "Alice has 3 more apples than Bob." *Inference rules* can be used to prove new logical forms from existing ones. Problems are constructed by sampling a *Proof Tree* with logical forms as nodes, leaves as axioms, and a question as the root, before programmatically converting nodes to natural language. See Appendix I.4 for details. MATHGAP includes only four logical forms with one being non-commutative (*transfer*, e.g. "Alice gave Bob 5 apples"). We extend MATHGAP in two ways:

- **Diversity:** We increase the diversity of logical forms and inference rules by adding nine statement types, six of which are non-commutative. Examples include "A eats 5 apples", "A and B switch the apples they have". Such statements make it more difficult for the model to keep track of the intermediate states before computing the final answer. See Appendix I.1 and I.5 for the full list and an example. Problem parameters are tree depth $d$ and inclusion of diverse logical forms.
- **Irrelevant statements:** We generate additional statements involving people irrelevant to the original problem and shuffle them into the original statements, such as "A is very generous and enjoys sharing food with others". See Appendix I.3 for the complete list. The problem parameters are the number of additional people and the number of additional sentences.

**Travel Planning** This task presents the model with a list of cities and various connecting modes of transit and asks the model to design a travel itinerary satisfying multiple constraints. This work is similar to that of Xie et al. (2024). For each task, we randomly generated a directed graph where the $S$ nodes represent cities and the edges represent connections. Each edge carries a subset of $A$ transportation modes, each with a randomly sampled cost. Based on this graph, we construct our travel planning problem, which consists of a word-based graph description and the constraints. The constraints include the starting and ending cities, a limit on the total travel cost, and $N$, the number of unique cities the traveler must visit. The problem parameters are $S$, $A$, and $N$. See Appendix J for further details.

## 4 RESULTS AND FAILURE ANALYSIS

Shortened problem descriptions, parameters, and evaluation results are shown in Figures 2, 3, 4, and 5. In most cases, increasing the "tediousness" of each task through the available parameters leads to a drop in performance. As expected, Gemma 3 has usually the weakest performance, the newest model, Gemini 3 Pro, is the best, and thinking models (o1, o3, R1, Gemini 2.5 and 3.0 Pro) typically outperform the non-thinking models (Gemma 3, Claude 3.7, Gemini 2.5 Flash). The GPT-5.1 model performs surprisingly bad in several cases – we suspect that this is due to the routing of problems to weaker models when the problem does not seem sufficiently complex, and due to this behavior we do not consider GPT-5.1 to be either a thinking or non-thinking model.

Overall, the results demonstrate that LLM performance scales poorly in problem parameters related to the amount of computation and storage, even on problems which are self-contained and quite easy for humans, and even for thinking models.

**Word Counting**: Given a text paragraph, count the occurrences of every word in a $k$-long list.
**Parameters**: number of words to count $k$, minimum paragraph size $m$.

| k | m | o1 | o3 | o5.1 | R1 | gem3 | C3.7 | G2.5F | G2.5P | G3 |
|---|-----|------|------|------|------|------|------|------|------|------|
| 1 | 50 | 1.00 | 0.95 | 0.80 | 0.95 | 0.85 | 0.85 | 0.80 | 0.95 | 1.00 |
|   | 150 | 0.95 | 1.00 | 0.40 | 0.90 | 0.45 | 0.55 | 0.65 | 0.80 | 1.00 |
| 3 | 50 | 1.00 | 0.95 | 0.75 | 0.90 | 0.55 | 0.65 | 0.50 | 0.80 | 1.00 |
|   | 150 | 0.65 | 1.00 | 0.25 | 0.55 | 0.05 | 0.20 | 0.15 | 0.85 | 1.00 |
| 6 | 50 | 0.95 | 0.95 | 0.50 | 0.95 | 0.30 | 0.60 | 0.35 | 0.80 | 1.00 |
|   | 150 | 0.70 | 1.00 | 0.15 | 0.35 | 0.00 | 0.25 | 0.00 | 0.70 | 0.80 |

**Character Counting**: Given a text paragraph, count the occurrences of a given character.
**Parameter**: minimum paragraph size $m$.

| m | o1 | o3 | o5.1 | R1 | gem3 | C3.7 | G2.5F | G2.5P | G3 |
|-----|------|------|------|------|------|------|------|------|------|
| 50 | 0.95 | 0.80 | 0.15 | 0.05 | 0.05 | 0.10 | 0.05 | 0.15 | 0.90 |
| 150 | 0.45 | 0.45 | 0.00 | 0.00 | 0.00 | 0.00 | 0.00 | 0.10 | 0.80 |

Figure 2: **Top**: The pass@5 performance on the word counting task vs. the number of words to count $k$ and minimum paragraph size $m$. While o1 performs well on word counting for the parameters in the table, it eventually fails with a sub $40\%$ accuracy with $k \geq 3$ and $m \geq 2000$. **Bottom:** The pass@5 performance for the single character counting task vs minimum paragraph size $m$.

**Logic Evaluation**: Given a propositional logic formula and four value assignments, identify which assignment evaluates to true.
**Parameters**: formula tree depth $d$, number of unique atomic propositions $n$.

| depth | sizes | o1 | o3 | o5.1 | R1 | gem3 | C3.7 | G2.5F | G2.5P | G3 |
|----|--------|------|------|------|------|------|------|------|------|------|
| 4 | medium | 1.00 | 1.00 | 0.97 | 1.00 | 0.95 | 0.98 | 0.90 | 1.00 | 1.00 |
|   | small | 0.98 | 1.00 | 0.97 | 0.98 | 0.93 | 0.97 | 0.97 | 1.00 | 1.00 |
| 8 | medium | 0.73 | 0.93 | 0.32 | 0.77 | 0.30 | 0.27 | 0.27 | 0.78 | 1.00 |
|   | small | 0.80 | 0.98 | 0.32 | 0.78 | 0.38 | 0.28 | 0.37 | 0.85 | 0.98 |
| 12 | medium | 0.35 | 0.45 | 0.37 | 0.35 | 0.33 | 0.35 | 0.32 | 0.52 | 0.53 |
|   | small | 0.43 | 0.33 | 0.33 | 0.38 | 0.25 | 0.38 | 0.33 | 0.55 | 0.53 |

**Logic Negation**: Given a propositional logic formula, identify its negation from four options.
**Parameters**: formula tree depth $d$, vocabulary for propositions, predicates, and domains.

| depth | values | o1 | o3 | o5.1 | R1 | gem3 | C3.7 | G2.5F | G2.5P | G3 |
|----|---------|------|------|------|------|------|------|------|------|------|
| 4 | letters | 0.95 | 1.00 | 0.95 | 0.15 | 0.85 | 0.95 | 0.85 | 1.00 | 1.00 |
|   | movies | 1.00 | 1.00 | 0.95 | 0.15 | 0.75 | 0.97 | 0.95 | 0.97 | 1.00 |
|   | rand 20 | 1.00 | 0.97 | 1.00 | 0.15 | 0.70 | 1.00 | 0.95 | 0.95 | 1.00 |
| 8 | letters | 0.97 | 1.00 | 0.93 | 0.17 | 0.60 | 0.93 | 0.88 | 1.00 | 1.00 |
|   | movies | 0.95 | 1.00 | 0.90 | 0.88 | 0.57 | 0.88 | 0.90 | 1.00 | 1.00 |
|   | rand 20 | 0.90 | 0.97 | 0.93 | 0.90 | 0.47 | 0.82 | 0.95 | 0.95 | 1.00 |
| 12 | letters | 0.75 | 0.88 | 0.85 | 0.82 | 0.45 | 0.80 | 0.82 | 0.93 | 0.88 |
|   | movies | 0.80 | 0.88 | 0.82 | 0.82 | 0.35 | 0.82 | 0.68 | 0.88 | 0.85 |
|   | rand 20 | 0.62 | 0.62 | 0.78 | 0.68 | 0.30 | 0.62 | 0.82 | 0.90 | 0.88 |

Figure 3: **Top**: Accuracy for the logic evaluation task vs. tree depth $d$ and number of possible unique predicates $n$. **Bottom**: Accuracy for the logic negation task vs. depth $d$ and the vocabulary used for propositions, predicates, and domains (random 20 denotes random character strings of length 20).

While it is difficult to pin down the causes of the model failures, we have performed an analysis of the failure symptoms evident from the answers and reasoning traces. We identified the following broad classes of errors:

- **Procedural errors**: the model makes an error in executing a simple step in a problem, such as performing arithmetic or a logical operation, which eventually leads to the wrong answer. These errors occur more often with increased "tediousness" in all tasks (paragraph length in the Counting, tree depth in Logic and ProofTree problems, and number of unique cities to visit in Travel Planning).

**ProofTree with diverse statements**: Given a diverse set of logical statements, answer questions that require deduction sampled from a proof tree with a bounded depth and number of leaves.
**Parameters**: max tree depth $d$, whether to include diverse logical forms, $\ell \in \{\text{True}, \text{False}\}$

| d | diverse | o1 | o3 | o5.1 | R1 | gem3 | C3.7 | G2.5F | G2.5P | G3 |
|---|---------|------|------|------|------|------|------|------|------|------|
| 3 | False | 1.00 | 0.70 | 0.85 | 1.00 | 0.95 | 0.90 | 1.00 | 1.00 | 1.00 |
|   | True  | 0.90 | 0.60 | 0.85 | 0.90 | 0.65 | 0.95 | 0.65 | 0.80 | 0.70 |
| 6 | False | 0.60 | 0.55 | 0.40 | 0.95 | 0.30 | 0.60 | 0.30 | 0.50 | 1.00 |
|   | True  | 0.70 | 0.90 | 0.45 | 0.65 | 0.15 | 0.55 | 0.30 | 0.45 | 0.65 |
| 9 | False | 0.35 | 0.75 | 0.15 | 0.55 | 0.15 | 0.20 | 0.30 | 0.25 | 0.80 |
|   | True  | 0.55 | 0.55 | 0.20 | 0.55 | 0.15 | 0.15 | 0.35 | 0.35 | 0.80 |

**ProofTree with irrelevant information**: Answer proof tree questions that include irrelevant information. **Parameters**: max tree depth $d$, number of irrelevant people $P$, number of irrelevant sentences $S$.

| P | S | o1 | o3 | o5.1 | R1 | gem3 | C3.7 | G2.5F | G2.5P | G3 |
|---|----|------|------|------|------|------|------|------|------|------|
| 1 | 0  | 0.50 | 0.60 | 0.50 | 0.75 | 0.35 | 0.50 | 0.25 | 0.60 | 1.00 |
|   | 60 | 0.45 | 0.45 | 0.25 | 0.50 | 0.25 | 0.20 | 0.10 | 0.50 | 0.80 |
| 2 | 0  | 0.50 | 0.55 | 0.45 | 0.75 | 0.20 | 0.55 | 0.20 | 0.50 | 0.95 |
|   | 60 | 0.45 | 0.40 | 0.25 | 0.45 | 0.15 | 0.15 | 0.20 | 0.40 | 0.75 |
| 4 | 0  | 0.40 | 0.60 | 0.25 | 0.65 | 0.15 | 0.20 | 0.35 | 0.35 | 0.90 |
|   | 60 | 0.25 | 0.40 | 0.10 | 0.25 | 0.00 | 0.10 | 0.10 | 0.35 | 0.80 |
| 6 | 0  | 0.40 | 0.60 | 0.30 | 0.55 | 0.00 | 0.15 | 0.20 | 0.30 | 0.95 |
|   | 60 | 0.30 | 0.30 | 0.05 | 0.35 | 0.05 | 0.00 | 0.05 | 0.35 | 0.80 |

Figure 4: Pass@5 scores for the proof tree tasks. **Top:** results for the diverse logic rules task, where we vary the depth $d$ and whether the diverse rules are included. **Bottom:** results for the irrelevant sentences task, where we vary $P$, the number of irrelevant people, and $S$, the number of irrelevant sentences.

**Travel Planning**: Create a travel itinerary using a city connection graph that adheres to a list of constraints. **Parameters**: num. cities $S$, num. transportation modes $A$, num. unique cities $N$

| S | steps | o1 | o3 | o5.1 | R1 | gem3 | C3.7 | G2.5F | G2.5P | G3 |
|----|----|------|------|------|------|------|------|------|------|------|
| 10 | 5 | 0.95 | 0.55 | 0.15 | 0.90 | 0.05 | 0.45 | 0.00 | 0.40 | 1.00 |
|    | 8 | 0.65 | 0.65 | 0.00 | 0.45 | 0.00 | 0.15 | 0.00 | 0.15 | 1.00 |
| 20 | 5 | 0.75 | 0.70 | 0.00 | 0.55 | 0.00 | 0.25 | 0.00 | 0.30 | 1.00 |
|    | 8 | 0.50 | 0.75 | 0.00 | 0.05 | 0.00 | 0.05 | 0.00 | 0.10 | 1.00 |

Figure 5: Travel planning: pass@5 performance results. We always have $A = 4$.

- **Omission**: The model misses or ignores a key step in the prompt. In ProofTree, models tend to ignore "transfer" operations, or key statements surrounded by irrelevant ones. In multi-word counting problems with long paragraphs, the models may miss relevant words.

- **Copying error**: The model copies text or values incorrectly, or incorrectly copies the reasoning outcome to the final answer.

- **Parsing**: The model fails to parse the question correctly; we observe that models can lose track of parentheses when reading a logic formula.

- **Hallucination**: The model hallucinates intermediate values or constraints. For example, in ProofTree problems, some models will hallucinate the initial number of items, or that married people have the same number of items. In Travel Planning, models sometimes propose a solution with hallucinated parameters which satisfies the constraints.

- **Shortcuts or heuristics**: Rather than executing computation, models sometimes prefer to exploit simplifications or take educated guesses, such as guessing the value of a logic formula based on its length.

- **Abandonment**: This is a special type of shortcut where the model concludes that the problem is too hard and refuses to answer. For example, for Travel Planning, models sometimes fail by randomly sampling a few solutions and concluding that the problem is infeasible. G2.5F is prone to giving up when evaluating long logic formulas.

- **Tokenization**: The error can be explained by difficulties in translation from words to tokens. For example, the character counting performance of all models is significantly lower than word counting, which suggests that tokenization is an issue for this task.

We attributed errors to these types using the following procedure, with a strong model (Gemini 3.0) as a grading assistant:

1. On a random subset of all incorrect responses, we prompted the grading assistant with the original question, the correct answer, and the incorrect solution, and asked for a summary of the errors made along with a single sentence identifying the primary cause of the error.

2. We used the same grading assistant to look at every summary sentence and, for every task separately, cluster the primary causes into the 20 most common types (10 in the case of word counting).

3. For every incorrect response, we prompted the grading assistant with the original question and answer, the incorrect solution, and the list of the 20 most common error types for the corresponding task. We then asked the model to list the errors in the solution and choose the common error that best represents the failure.

4. Finally, we clustered the representative errors found in the previous step, by hand, into the categories introduced above.

The results for several models (not including Gemini 3.0 Pro) are presented in Figure 6, analyzing the error cases in the above tasks (except for the least interesting character counting task); more detailed results and details of the prompting strategy for the analysis are given in Appendix C. Broadly, we found that all models make procedural errors, as well as omission errors. They also hallucinate quite a lot when the problem involves a composition of reasoning and natural language (ProofTree and Travel Planning) while hallucination is less of a problem in the clean logic and and counting tasks.

When comparing thinking and non-thinking models, we observe that thinking models are less prone to shortcuts, which demonstrates the real strength of producing reasoning traces, and they seem to commit fewer omission errors (except for Gemini 2.5 Flash in the logic evaluation problems, where these may be masked by other types of errors, such as abandonment). Otherwise, the error types varied across models, and seem to be specific to a model's performance on a given task (e.g., o3 usually makes very few procedural errors, except for the logic negation problem, while R1 makes several procedure errors in the logic problems, suggesting that such problems were not emphasized in its training data). Finally, perhaps surprisingly, we do not always find that thinking models make fewer procedural errors than non-thinking models, for example, Gemini 2.5. Flash makes fewer procedural errors than o1 and o3 in the logic tasks, but a proper comparison is hard due to the different (and undisclosed) parameters of the models.

One would expect that poor out-of-distribution generalization and too much reliance on memorization should also be problematic for LLMs. We see some evidence for this; for example, changing the vocabulary in the logic tasks highlights errors due to poor out-of-distribution generalization: generally, the performance is the best when the problem variables are single letters (which is likely the format of logical problems in the training data), and worst for random 20-character strings. We can also attribute some of the ProofTree failures to poor out-of-distribution generalization, as some of the introduced statements (such as "A and B switch their apples") are not common in math word problems. Nevertheless, without knowing exactly what is in the training data, it is hard to argue how much the models rely on memorization, and it is hard to attribute any of the error-types above to memorization (some hallucination errors may be due to memorization, such as assuming that married couples have the same properties). To examine this issue better, in the next section we present a problem type, consisting of trivialized versions of well-known logic puzzles, where we have good reasons to assume that similar problems are in the training data of the models and we can show that the errors made by the models are the direct consequence of relying too much on similar memorized data.

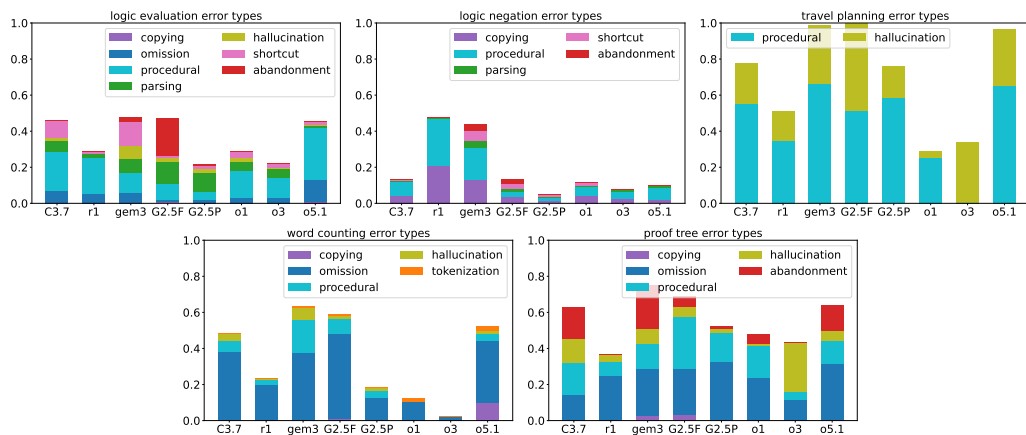

Figure 6: Breakdown of error types for each model and each task.

| Model | G1.5 | G2.0F | G2.5P | gem3 | C3.5 | C3.7 | 4o | o1 | o3 | R1 |
|---|---|---|---|---|---|---|---|---|---|---|
| Puzzle | 79.4 | 78.4 | 93.8 | 68.0 | 63.9 | 77.3 | 75.3 | 86.7 | 87.6 | 87.6 |
| Unpuzzle | 17.5 | 38.1 | 62.9 | 34.0 | 27.8 | 48.5 | 19.6 | 59.8 | 74.2 | 59.8 |

Table 1: Percentage of correct answers on puzzles and unpuzzles.

| Model | G1.5 | G2.0F | G2.5P | gem3 | C3.5 | C3.7 | 4o | o1 |
|---|---|---|---|---|---|---|---|---|
| Context corruption (CC) | 80 | 59 | 34 | 56 | 63 | 41 | 76 | 38 |
| CC, correct | 7 | 6 | 4 | 2 | 12 | 4 | 13 | 6 |
| CC, incorrect with delirium | 40 | 36 | 20 | 25 | 26 | 14 | 31 | 22 |
| CC, incorrect (other) | 33 | 16 | 10 | 29 | 25 | 23 | 32 | 10 |

Table 2: Number of unpuzzle solutions (out of 97) containing "context corruption." We further subcategorize corrupt solutions as (i) correct: leading to a correct final answer; (ii) incorrect with delirium: leading to an incorrect final answer with a solution that corresponds nearly exactly to the solution of the original puzzle; (iii) incorrect (other): leading to an incorrect final answer for other reasons. R1 and o3 are omitted since the answers we obtained often did not include full reasoning.

## 5 UNPUZZLES

To examine the aforementioned memorization problem, we introduce the UNPUZZLES dataset, which consists of 97 well-known logical puzzles that are commonly found on the internet, and their *trivialized* versions, manually constructed by formulating textually similar questions that remove difficulty. While the puzzles typically require reasoning and background math knowledge, the answers to the unpuzzles are intended to be immediately obvious by common sense. See Appendix D for more details, dataset creation instructions, and some examples. As we will show, all language models perform much better on the puzzles than on the unpuzzles, suggesting that they rely on memorized input patterns to generate answers rather than performing true logical reasoning.

To provide further evidence of memorization, we created a dataset of *context-shifted* (CS) unpuzzles. Each CS unpuzzle is textually different from the corresponding unpuzzle but retains the same logical structure; that is, its answer is equally obvious. Performing poorly on an unpuzzle and well on a corresponding CS unpuzzle would suggest that the failure is due to the memorization of the puzzle text and solution, rather than inherent inability to reason about the problem. We generated CS unpuzzles automatically for a subset of 64 unpuzzles with simple numerical or categorical answers (as opposed to, e.g., puzzles asking for a strategy). We prompted models (o1 and Gemini 2.0 Flash) to rewrite each unpuzzle and change the language and setting, but keep the same logical structure and answer. We then verified and optionally edited the results. See Appendix D.2 for details. Figure 1 shows an example of a puzzle, unpuzzle, and CS unpuzzle.

| Model | G2.0F | G2.0FT | G2.5P | gem3 | C3.7 | 4o | o1 | o3 | R1 |
|---|---|---|---|---|---|---|---|---|---|
| Puzzle Score | 67 | 66 | 72 | 52 | 67 | 58 | 77 | 73 | 80 |
| Unpuzzle Score | 53 | 36 | 55 | 41 | 55 | 33 | 50 | 75 | 67 |
| Context-shifted Score | 70 | 48 | 66 | 50 | 63 | 52 | 59 | 80 | 73 |

Table 3: Percentage accuracy of all tested models on the original puzzles, the unpuzzles, and the context-shifted unpuzzles, for a subset of 64 problems with numerical or categorical answers. We see that every model performs better on the context-shifted unpuzzles than the unpuzzles, indicating that similarity to the puzzles degrades performance. In some cases, performance on the context-shifted unpuzzles is higher than for the original puzzles.

---

**o1 solution to the Chameleons Go on a Date unpuzzle (excerpt)**

A succinct way to see that they can all end up the same color (and in fact which color that must be) is via the following "invariant" argument: **[...]**
Define the quantity S = P + 2Y (mod 3). **[...]**
Since the initial S = 0(mod 3), the only way to remain consistent with that invariant and end up with all chameleons of one hue is if they all become Maroon (because only that final state also gives S = 0).
Conclusion: Yes. It is possible for all of the chameleons to end up the same color; specifically, they can all turn Maroon.

---

Figure 7: Chameleons go on a date (Figure 1): solution by o1. The answer to the original puzzle is 'no' through a proof by contradiction. The answer to the unpuzzle is 'yes'. o1 provides the correct final answer, but an incorrect solution; it finds that the contradiction does not hold and answers 'yes' without a constructive proof.

### 5.1 EVALUATION

**Correctness** We generated the solution to each puzzle and unpuzzle independently using each model. We first verified whether the final answer to each is correct or not (regardless of whether the solution leading to the answer is correct). The evaluation was performed manually by four human annotators, since the answers to some puzzles are strategies rather than simple values. Each answer was assessed by a single annotator, or by consensus of all annotators if marked ambiguous.

**Context corruption** Next, we characterize the extent to which the poor performance on the unpuzzles is a consequence of memorization of the original puzzles. We define "context corruption" in an unpuzzle solution as erroneous or superfluous content (e.g. assumptions or reasoning steps) inappropriately recalled from the original puzzle or its solution. We evaluated each unpuzzle solution according to whether it contains context corruption or not. The most extreme behavior is when the models provided a solution that is nearly identical to the puzzle solution, sometimes without acknowledging that the unpuzzle is different – we call this category "delirium." We omit o3 and R1 as they often just responded with the final answer, making the degree of context corruption unclear, though their erroneous answers usually correspond to the answer to the original puzzle. This evaluation was performed by four human annotators and summarized in Table 2. We observe that memorization artifacts from the original puzzle and its solution are found in most cases, and even thinking models simply output the solution to the original puzzle about a fifth of the time. See Figure 7 for an example of context corruption in o1's solution to "Chameleons Go on a Date." See Appendix E for more illustrative examples as well as some amusing answers.

**Context-shifted evaluation** We evaluated models on the size-64 subset with corresponding CS unpuzzles. The results are shown in Table 3. We note that all models perform better on the context-shifted version of the dataset, which offers further evidence that the poor performance on the unpuzzles is due to the wording (and memorization of the original puzzles), rather than models' inherent inability to reason about the problems.

The UNPUZZLES dataset complements the procedural evaluations by providing another benchmark that is easy for humans and difficult for LLMs. It illustrates that the good performance of the models on the original (difficult) puzzles is at least in part a consequence of memorization of internet data, rather than true problem-solving abilities.

## 6 DISCUSSION

In a society that is increasingly utilizing frontier language models, understanding the capabilities and weaknesses of these models is becoming more and more important. We have presented a comprehensive set of procedurally-generated parametric problems that are inherently easy (if tedious) for humans, and designed to assess LLM failures due to statistical shortcut learning, procedural errors or hallucinations due to long context and long reasoning chains. As we demonstrate, these problems can be made difficult enough to make all SOTA LLMs fail. One suggestion from our paper is that LLMs should be evaluated not only by the most difficult problem they can solve, but also by the simplest problem they struggle with.

Our procedural problems also suggest that some errors are due to relying on memorized patterns instead of performing proper reasoning. To investigate this problem, we have provided a small human-curated UNPUZZLES dataset of trivialized versions of math and logic puzzles commonly found on the internet. Our analysis shows that all models perform significantly worse on the unpuzzles than on the original puzzles, in most cases due to memorization of web data. This demonstrates that oftentimes LLMs mimic training data rather than performing true reasoning, making it relatively easy to find out-of-distribution problems where the models fail, and this problem is also present at the newest thinking models (while similar conclusions were hypothesized in other recent works, our result is the first to show this without actual access to the training data). This suggests that users remain careful when relying on the output of LLMs.

The main limitation of our work is that most of the experiments were run on closed-source models, which limits our ability to understand shortcomings beyond observing trends in the experiments and inspecting reasoning traces when available. We hope that our benchmarks will be useful in assessing and improving the reasoning capabilities of future generations of models.

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

# A  LICENCES FOR EXISTING ASSETS

## A.1  MODELS

Below, we've tabulated the specific models and licences we have used.

OpenAI The specific models we used from OpenAI are `gpt-4o-2024-08-06`, `o1-2024-12-17`, and `o3-2025-04-16`, which were abbreviated by 4o, o1, and o3 in the text. Terms of Use can be found at `https://openai.com/policies/row-terms-of-use/`.

Anthropic We used `claude-3-5-sonnet-20240620` and `claude-3-7-sonnet-20250219`, which were abbreviated C3.5 and C3.7. Terms of Service can be found at `https://privacy.anthropic.com/en/articles/9190861-terms-of-service-updates`.

Gemma 3: We used the 27b-it model, which has open weights and permits responsible commercial use. Terms of Service are given at `https://gemma3.app/terms-of-service`.

DeepSeek DeepSeek's R1 model and weights are licenced under the MIT licence DeepSeek-AI (2025).

Gemini The Gemini 2.0 Flash, 2.0 Flash Thinking, and 2.5 Pro had API names of `gemini-2.0-flash-001`, `gemini-2.0-flash-thinking-exp` and `gemini-2.5-pro-exp-03-25`. Terms of service can be found at `https://ai.google.dev/gemini-api/terms`.

## A.2  DATA

We list the websites used to collect math and logic puzzles and their licences and terms of use below. Please see the released dataset for per-puzzle attributions.

- Wikipedia (`https://www.wikipedia.org/`): CC BY-SA 4.0 Creative Commons Attribution-ShareAlike 4.0 International `https://creativecommons.org/licenses/by-sa/4.0/`
- `www.mathisfun.com` copyright Rod Pierce, cited as instructed on the website (Pierce)
- `https://puzzles.nigelcoldwell.co.uk/` copyright Nigel Coldwell.
- `https://geeksforgeeks.org/`, Terms of Use `https://www.geeksforgeeks.org/legal/intellectual-property-rights-legal/`

# B  PROCEDURAL LOGIC RESULTS WITH CONFIDENCE INTERVALS

We now include the procedural logic results including simple Gaussian error bars; these were omitted from the main body due to space constrains. In particular, results for OpenAI's o3 model are included. You can find the results in Tables 4-10.

| m | o1 | o3 | 5.1 | R1 | gem3 | C3.7 | G2.5F | G2.5P | G3 |
|---|---|---|---|---|---|---|---|---|---|
| 50 | 0.95±0.10 | 0.80±0.18 | 0.15±0.16 | 0.05±0.10 | 0.05±0.10 | 0.10±0.14 | 0.05±0.10 | 0.15±0.16 | 0.90±0.14 |
| 150 | 0.45±0.23 | 0.45±0.23 | 0.00±0.00 | 0.00±0.00 | 0.00±0.00 | 0.00±0.00 | 0.00±0.00 | 0.10±0.14 | 0.80±0.18 |

Table 4: Full results, with confidence intervals, for the Character Count task

# C  AUTO-GRADING THE ERRORS IN RESPONSES

This section provides more details about the auto-grading. In the main paper, we described four steps for auto-grading the responses; we elaborate on each below. This process was repeated for the word count, logic evaluation, logic negation, ProofTree, and travel planning tasks. The errors in the

| k | m | o1 | o3 | o5.1 | R1 | gem3 | C3.7 | G2.5F | G2.5P | G3 |
|---|---|----|----|------|----|------|------|-------|-------|----|
| 1 | 50 | 1.00±0.00 | 0.95±0.10 | 0.80±0.18 | 0.95±0.10 | 0.85±0.16 | 0.85±0.16 | 0.80±0.18 | 0.95±0.10 | 1.00±0.00 |
|   | 150 | 0.95±0.10 | 1.00±0.00 | 0.40±0.22 | 0.90±0.14 | 0.45±0.23 | 0.55±0.23 | 0.65±0.22 | 0.80±0.18 | 1.00±0.00 |
| 3 | 50 | 1.00±0.00 | 0.95±0.10 | 0.75±0.20 | 0.90±0.14 | 0.55±0.23 | 0.65±0.22 | 0.50±0.23 | 0.80±0.18 | 1.00±0.00 |
|   | 150 | 0.65±0.22 | 1.00±0.00 | 0.25±0.20 | 0.55±0.23 | 0.05±0.10 | 0.20±0.18 | 0.15±0.16 | 0.85±0.16 | 1.00±0.00 |
| 6 | 50 | 0.95±0.10 | 0.95±0.10 | 0.50±0.23 | 0.95±0.10 | 0.30±0.21 | 0.60±0.22 | 0.35±0.22 | 0.80±0.18 | 1.00±0.00 |
|   | 150 | 0.70±0.21 | 1.00±0.00 | 0.15±0.16 | 0.35±0.22 | 0.00±0.00 | 0.25±0.20 | 0.00±0.00 | 0.70±0.21 | 0.80±0.18 |

Table 5: Full results, with confidence intervals, for the Word Count task

| d | sizes | o1 | o3 | o5.1 | R1 | gem3 | C3.7 | G2.5F | G2.5P | G3 |
|---|-------|----|----|------|----|------|------|-------|-------|----|
| 4 | medium | 1.00±0.00 | 1.00±0.00 | 0.97±0.05 | 1.00±0.00 | 0.95±0.06 | 0.98±0.03 | 0.90±0.08 | 1.00±0.00 | 1.00±0.00 |
|   | small | 0.98±0.03 | 1.00±0.00 | 0.97±0.05 | 0.98±0.03 | 0.93±0.06 | 0.97±0.05 | 0.97±0.05 | 1.00±0.00 | 1.00±0.00 |
| 8 | medium | 0.73±0.12 | 0.93±0.06 | 0.32±0.12 | 0.77±0.11 | 0.30±0.12 | 0.27±0.12 | 0.27±0.12 | 0.78±0.11 | 1.00±0.00 |
|   | small | 0.80±0.10 | 0.98±0.03 | 0.32±0.12 | 0.78±0.11 | 0.38±0.13 | 0.28±0.12 | 0.37±0.13 | 0.85±0.09 | 0.98±0.03 |
| 12 | medium | 0.35±0.12 | 0.45±0.13 | 0.37±0.13 | 0.35±0.12 | 0.33±0.12 | 0.35±0.12 | 0.32±0.12 | 0.52±0.13 | 0.53±0.13 |
|   | small | 0.43±0.13 | 0.33±0.12 | 0.33±0.12 | 0.38±0.13 | 0.25±0.11 | 0.38±0.13 | 0.33±0.12 | 0.55±0.13 | 0.53±0.13 |

Table 6: Full results, with confidence intervals, for the Logic Evaluation task

character counting were essentially all due to tokenization and memorizing the number of different characters in each token, and therefore less interesting.

1. On a sampled subset of all incorrect responses, we prompted the grading assistant with the original question, the correct answer, and the incorrect solution and asked for a summary of the errors made along with a single sentence identifying the primary cause of the error:

---

**Auto-grading step 1 template**

```
You are an expert at identifying errors in solutions to logic
and reasoning problems. I will give you such a problem, the correct answer,
then an incorrect solution. I want you to analyze the solution and find
the errors that lead to the incorrect answer.

The question is: {question}.
End question.

The correct answer is: {correct_answer}.

The incorrect response is: {incorrect_response}.
End response.

Please list specific errors in the response that contribute to the
incorrect solution. Then, in a single sentence, describe the primary
reason for the incorrect answer.
```

---

2. We used the same grading assistant to look at every summary sentence and, for every task separately, cluster the primary causes into the 20 most common types (for word counting we use only 10 types due to the simplicity of the task).

| d | values | o1 | o3 | o5.1 | R1 | gem3 | C3.7 | G2.5F | G2.5P | G3 |
|---|---|---|---|---|---|---|---|---|---|---|
| 4 | letters | 0.95±0.07 | 1.00±0.00 | 0.95±0.07 | 0.15±0.11 | 0.85±0.11 | 0.95±0.07 | 0.85±0.11 | 1.00±0.00 | 1.00±0.00 |
|   | movies | 1.00±0.00 | 1.00±0.00 | 0.95±0.07 | 0.15±0.11 | 0.75±0.14 | 0.97±0.05 | 0.95±0.07 | 0.97±0.05 | 1.00±0.00 |
|   | rand20 | 1.00±0.00 | 0.97±0.05 | 1.00±0.00 | 0.15±0.11 | 0.70±0.15 | 1.00±0.00 | 0.95±0.07 | 0.95±0.07 | 1.00±0.00 |
| 8 | letters | 0.97±0.05 | 1.00±0.00 | 0.93±0.08 | 0.17±0.12 | 0.60±0.16 | 0.93±0.08 | 0.88±0.11 | 1.00±0.00 | 1.00±0.00 |
|   | movies | 0.95±0.07 | 1.00±0.00 | 0.90±0.10 | 0.88±0.11 | 0.57±0.16 | 0.88±0.11 | 0.90±0.10 | 1.00±0.00 | 1.00±0.00 |
|   | rand20 | 0.90±0.10 | 0.97±0.05 | 0.93±0.08 | 0.90±0.10 | 0.47±0.16 | 0.82±0.12 | 0.95±0.07 | 0.95±0.07 | 1.00±0.00 |
| 12 | letters | 0.75±0.14 | 0.88±0.11 | 0.85±0.11 | 0.82±0.12 | 0.45±0.16 | 0.80±0.13 | 0.82±0.12 | 0.93±0.08 | 0.88±0.11 |
|   | movies | 0.80±0.13 | 0.88±0.11 | 0.82±0.12 | 0.82±0.12 | 0.35±0.15 | 0.82±0.12 | 0.68±0.15 | 0.88±0.11 | 0.85±0.11 |
|   | rand20 | 0.62±0.16 | 0.62±0.16 | 0.78±0.13 | 0.68±0.15 | 0.30±0.15 | 0.62±0.16 | 0.82±0.12 | 0.90±0.10 | 0.88±0.11 |

Table 7: Full results, with confidence intervals, for the Logic Negation task

| d | diverse | o1 | o3 | o5.1 | R1 | gem3 | C3.7 | G2.5F | G2.5P | G3 |
|---|---|---|---|---|---|---|---|---|---|---|
| 3 | False | 1.00±0.00 | 0.70±0.21 | 0.85±0.16 | 1.00±0.00 | 0.95±0.10 | 0.90±0.14 | 1.00±0.00 | 1.00±0.00 | 1.00±0.00 |
|   | True | 0.90±0.14 | 0.60±0.22 | 0.85±0.16 | 0.90±0.14 | 0.65±0.22 | 0.95±0.10 | 0.65±0.22 | 0.80±0.18 | 0.70±0.21 |
| 6 | False | 0.60±0.22 | 0.55±0.23 | 0.40±0.22 | 0.95±0.10 | 0.30±0.21 | 0.60±0.22 | 0.30±0.21 | 0.50±0.23 | 1.00±0.00 |
|   | True | 0.70±0.21 | 0.90±0.14 | 0.45±0.23 | 0.65±0.22 | 0.15±0.16 | 0.55±0.23 | 0.30±0.21 | 0.45±0.23 | 0.65±0.22 |
| 9 | False | 0.35±0.22 | 0.75±0.20 | 0.15±0.16 | 0.55±0.23 | 0.15±0.16 | 0.20±0.18 | 0.30±0.21 | 0.25±0.20 | 0.80±0.18 |
|   | True | 0.55±0.23 | 0.55±0.23 | 0.20±0.18 | 0.55±0.23 | 0.15±0.16 | 0.15±0.16 | 0.35±0.22 | 0.35±0.22 | 0.80±0.18 |

Table 8: Full results, with confidence intervals, for the MathGap Diverse task

---

**Auto-grading step 1 template**

```
You are an expert instructor for logic problems and familiar with
mathematical word problems. I have a collection of word problems with
incorrect solutions. I have already gone through them and listed the
errors as well as summary sentences for the primary reason the solution
was wrong.

Please go through all the examples and provide the twenty most common mistakes
in the incorrect solutions.
{concatenated responses from part one}
```

3. For every incorrect response, we prompted the grading assistant with the original question and answer, the incorrect solution, and the list of 20 most common error for the corresponding task. We then asked the model to list the errors in the solution and choose the common error that best represents the failure.

| P | S | o1 | o3 | o5.1 | R1 | gem3 | C3.7 | G2.5F | G2.5 | G3 |
|---|---|----|----|------|----|------|------|-------|------|----|
| 1 | 0 | 0.50±0.23 | 0.60±0.22 | 0.50±0.23 | 0.75±0.20 | 0.35±0.22 | 0.50±0.23 | 0.25±0.20 | 0.60±0.22 | 1.00±0.00 |
|   | 60 | 0.45±0.23 | 0.45±0.23 | 0.25±0.20 | 0.50±0.23 | 0.25±0.20 | 0.20±0.18 | 0.10±0.14 | 0.50±0.23 | 0.80±0.18 |
| 2 | 0 | 0.50±0.23 | 0.55±0.23 | 0.45±0.23 | 0.75±0.20 | 0.20±0.18 | 0.55±0.23 | 0.20±0.18 | 0.50±0.23 | 0.95±0.10 |
|   | 60 | 0.45±0.23 | 0.40±0.22 | 0.25±0.20 | 0.45±0.23 | 0.15±0.16 | 0.15±0.16 | 0.20±0.18 | 0.40±0.22 | 0.75±0.20 |
| 4 | 0 | 0.40±0.22 | 0.60±0.22 | 0.25±0.20 | 0.65±0.22 | 0.15±0.16 | 0.20±0.18 | 0.35±0.22 | 0.35±0.22 | 0.90±0.14 |
|   | 60 | 0.25±0.20 | 0.40±0.22 | 0.10±0.14 | 0.25±0.20 | 0.00±0.00 | 0.10±0.14 | 0.10±0.14 | 0.35±0.22 | 0.80±0.18 |
| 6 | 0 | 0.40±0.22 | 0.60±0.22 | 0.30±0.21 | 0.55±0.23 | 0.00±0.00 | 0.15±0.16 | 0.20±0.18 | 0.30±0.21 | 0.95±0.10 |
|   | 60 | 0.30±0.21 | 0.30±0.21 | 0.05±0.10 | 0.35±0.22 | 0.05±0.10 | 0.00±0.00 | 0.05±0.10 | 0.35±0.22 | 0.80±0.18 |

Table 9: Full results, with confidence intervals, for the MathGap irrelevant tasks

| S | steps | o1 | o3 | o5.1 | R1 | gem3 | C3.7 | G2.5F | G2.5P | G3 |
|---|-------|----|----|------|----|------|------|-------|-------|----|
| 10 | 5 | 0.95±0.10 | 0.55±0.23 | 0.15±0.16 | 0.90±0.14 | 0.05±0.10 | 0.45±0.23 | 0.00±0.00 | 0.40±0.22 | 1.00±0.00 |
|    | 8 | 0.65±0.22 | 0.65±0.22 | 0.00±0.00 | 0.45±0.23 | 0.00±0.00 | 0.15±0.16 | 0.00±0.00 | 0.15±0.16 | 1.00±0.00 |
| 20 | 5 | 0.75±0.20 | 0.70±0.21 | 0.00±0.00 | 0.55±0.23 | 0.00±0.00 | 0.25±0.20 | 0.00±0.00 | 0.30±0.21 | 1.00±0.00 |
|    | 8 | 0.50±0.23 | 0.75±0.20 | 0.00±0.00 | 0.05±0.10 | 0.00±0.00 | 0.05±0.10 | 0.00±0.00 | 0.10±0.14 | 1.00±0.00 |

Table 10: Full results, with confidence intervals, for the Travel task

```
Auto-grading step 1 template

You are an expert at identifying errors in solutions to logic and
mathematical word problems and an excellent tutor. I will give you
a question that can be solved with simple logical reasoning
followed by an incorrect response to that question.

I want you to first identify the mistakes in the incorrect response
then describe the primary causes of the incorrect answer

The question is: {question}.
End of question.

The correct answer is: {answer}.

The incorrect response is: {response}.
End of response.

From the follow twenty options, please find the error that contributed
the most to the incorrect answer and point out specifically where the
error was made.
{list of mistakes as below}
Please point out where the primary error is and answer using the
template
Final answer: 1 or 2 or ...or 20.
```

4. Finally, we extracted the error per question and clusters the errors, by hand, into the nine main categories.

## C.1 ERROR ANALYSIS FOR WORD COUNTING

The following table introduces the ten most common failure modes, as well an the main error category they fall into, for the word counting task. The biggest error for every model was a failure to count repeated words in close proximity. Some other noteworthy anomalies are Gemma 3's confusion about matching plurals and Gemini 2.5 pro's failures to count words towards the end of the paragraph.

1. **Inexact String Matching (Singular/Plural Conflation)** (procedure): Counting "markets" when the target is "market," "Koreans" for "Korean," or "statues" for "statue." Conversely, counting the singular "tank" when the target is the plural "tanks."

2. **Repetition Blindness (Intra-Sentence Proximity)** (omission): In phrases like "tracking from a few feet away from" or "experimented with hollow shot filled with," the solution identifies the first instance but skips the second.

3. **End-of-Text Attentional Decay** (omission): The solution counts correctly through the first 80% of the text but misses words in phrases like "pressure from his fellow legislators" or "candidates that included..." appearing at the very end.

4. **Failure to Parse Dense Clusters** (omission): Missing instances of "were" in a sentence where the word appears 6 or 7 times (e.g., "were sent... were assigned... were destroyed").

5. **Internal Inconsistency (Reasoning vs. Output)** (copying): The "Step-by-Step" section correctly identifies 3 instances of a word, but the final output list records the number 4 or 2. Or, the solver lists instances but sums them incorrectly.

6. **Repeated Phrase Oversight** (omission): If the phrase "center of government" appears twice in the text, the solver often counts it as one occurrence, failing to realize that the phrase (and the target word "government" inside it) actually occurs twice at different locations.

7. **Function Word Blindness** (omission): Consistently undercounting high-frequency words like "that," "with," "were," and "from."

8. **Hallucination and Double-Counting** (hallucination): Counting a word because it *should* be there based on context (e.g., in a quote not actually present in the text) or identifying "shot camera" and "camera tracking" as two separate occurrences of "camera" when they are actually the same word in the text "shot camera tracking."

9. **Tokenization and Punctuation Failures** (tokenization): Missing "wooden" in "wooden@-@unk" because of the hyphen, or missing "tank" in "tank's" because of the apostrophe.

10. **Semantic Filtering (False Exclusions)** (procedure): Excluding the word "Hill" because it refers to a person's name (proper noun) rather than a geographic location, or excluding "infantry" because it appears as a general adjective rather than part of a specific unit title (e.g., "2nd Infantry Division").

## C.2 ERROR ANALYSIS FOR LOGIC EVALUATION

As before, we introduce the twenty most common errors and the error breakdown per model. A surprisingly common failure mode was disregarding a global negation. Another source of error was misidentifying the overall-logical structure, through a parenthesis parsing error or otherwise. Surprisingly, Gemini 2.5 pro had the largest number of abandonment errors, though one might argue that refusing to solve the problem is a more desirable outcome than confidently providing an incorrect answer.

1. **Failure to Identify Vacuous Truth (Implication Shortcuts)** (procedure): Failing to recognize that if the antecedent ($P$) of an implication ($P \rightarrow Q$) is False, the entire implication is automatically True, regardless of the complexity of the consequent ($Q$).

2. **Overlooking Disjunction Short-Circuiting** (procedure) Failing to recognize that if the first operand ($P$) of a disjunction ($P \vee Q$) is True, the entire expression is True, regardless of the second operand.

3. **Misidentification of the Main Logical Connective** (parsing): Incorrectly parsing the top-level structure of the formula.

4. **Incorrect Scope of Negation** (parsing): Misinterpreting which part of the formula a 'not' operator applies to.

5. **Operator Precedence Failures** (procedure): Evaluating logical operators in the wrong order when parentheses are not explicit, or misinterpreting standard precedence rules.

6. **Biconditional Truth Value Logic Error** (procedure): Assuming that 'False' $\leftrightarrow$ 'False' evaluates to False.

7. **One-Sided Biconditional Evaluation** (procedure): Proving that the Left-Hand Side (LHS) of a biconditional ($P \leftrightarrow Q$) is True and immediately concluding the formula is True.

8. **Implication Calculation Error (True $\rightarrow$ False)** (procedure): Evaluating an implication with a True Antecedent and a False Consequent as True.

9. **Conjunction Block Failure** (procedure): Failing to notice that a single False term in a top-level Conjunction ($P \wedge Q \wedge R$) invalidates the entire formula.

10. **Premature Termination (Partial Evaluation)** (shortcut): Evaluating only a fragment of the formula (e.g., the first sub-clause) and assuming it dictates the final answer.

11. **Reliance on Invalid Heuristics** (shortcut): Using "rules of thumb" instead of boolean algebra.

12. **Refusal to Solve (Abandonment)** (abandonment): Explicitly giving up on the evaluation due to complexity.

13. **Hallucination of External Tool Verification** (hallucination): Claiming to have used a Python script or "symbolic logic solver" to verify an answer that is objectively incorrect.

14. **Variable Assignment Errors** (copying): Using the truth values from one option (e.g., Option A) while evaluating another option (e.g., Option C).

15. **Necessity vs. Sufficiency Confusion** (procedure): Treating a sub-expression as a necessary condition when it is not.

16. **Neglecting the Global Negation** (omission): Correctly evaluating the massive inner formula but forgetting to apply the outermost 'not(...)'.

17. **Incorrect Parsing of Parentheses Depth** (parsing): Losing track of nesting depth, often "closing" a parenthesis too early or too late.

18. **"False implies False" Confusion** (procedure): Believing that 'False' $\rightarrow$ 'False' evaluates to False.

19. **Ignoring Biconditional Mismatches** (procedure): Failing to spot that $True \leftrightarrow False$ evaluates to False.

20. **False Generalization from Partial Data** (shortcut): Analyzing one option, finding a specific sub-structure behaves a certain way, and assuming that behavior applies to all other options without verification.

## C.3 ERROR ANALYSIS FOR LOGIC NEGATION

The logic negation errors show some interesting patters. For example, R1 and Gemma 3 are both prone to incorrectly copying predicates or variables, whereas the other models are not. Double negation is a common failure mode across most models; it seems like negation is generally difficult. There are some specific failures: R1 does not know how to take a biconditional expansion, and Gemma 3 frequently used a visual difference heuristic no other mode

1. **Double Negation Oversight** ($\neg(\neg P) \equiv P$) (procedure): The most frequent error. When asked to negate a statement that already begins with a negation (e.g., 'not (Exists x...)'), solvers often attempt to distribute a new negation into the inner formula (changing quantifiers and connectives) rather than recognizing that the correct answer is simply the inner statement with the outer "not" removed.

2. **Corruption of the Antecedent in Implications** (procedure): When negating a conditional statement ($P \rightarrow Q$), the correct negation is $P \wedge \neg Q$. A very common mistake is to negate or modify the antecedent $P$ (e.g., flipping quantifiers inside $P$ or adding a "not"), failing to realize that the antecedent must remain exactly identical to the original.

3. **Retaining the Implication Operator** (procedure): Solvers often fail to change the main operator from an implication ($\rightarrow$) to a conjunction ($\wedge$) during negation. They incorrectly produce a statement like $P \vee \neg Q$ or $P \wedge Q$ instead of the required $P \wedge \neg Q$.

4. **Recursive Biconditional Negation** (procedure): When negating an equivalence ($P$ iff $Q$), solvers frequently assume the negation is $\neg P$ iff $\neg Q$ (negating both sides recursively). This results in a logically equivalent statement, not a negation. The correct negation is the exclusive disjunction ($P \wedge \neg Q) \vee (\neg P \wedge Q)$.

5. **Incomplete Biconditional Expansion** (procedure): Even when solvers recognize that a negated biconditional requires an XOR structure, they often fail to preserve the un-negated sides correctly. For example, they might produce ($P \wedge \neg Q) \vee (\neg P \wedge \neg Q)$ or fail to keep $P$ identical in the first disjunct.

6. **De Morgan's Law Failures** (procedure): When negating conjunctions or disjunctions, solvers often flip the operator (AND ↔ OR) but fail to negate the individual terms, or conversely, negate the terms but fail to flip the operator.

7. **Quantifier Inversion in Non-Negated Scopes** (copying): Solvers frequently flip quantifiers ($\forall \leftrightarrow \exists$) universally throughout the entire formula, including within sub-formulas that should be preserved (such as the antecedent of a negated implication or the un-negated side of a biconditional expansion).

8. **Failure to Negate Quantifiers** (procedure): Conversely, in sections that *should* be negated, solvers often negate the predicates but forget to swap the quantifier (e.g., leaving "Exists" as "Exists" while negating the inner proposition).

9. **Misidentification of the Main Connective** (procedure): Solvers often misidentify the top-level logical operator. For example, analyzing a complex statement as a "Conjunction" when the main operator is actually an "Implication" or "Biconditional," leading to the wrong negation strategy.

10. **Parsing Scope Errors** (parsing): Misinterpreting the scope of parentheses or quantifiers. A common error is assuming a quantifier applies only to the immediate next term, when it actually scopes over a subsequent biconditional or implication (e.g., negating $\exists x(P \to Q)$ as $\forall x P \to \neg Q$ instead of $\forall x(P \land \neg Q)$).

11. **Negating "A and B" as an Implication** (procedure): Solvers sometimes incorrectly negate a conjunction $A \land B$ using an implication structure like $\neg A \to \neg B$, rather than the correct De Morgan's expansion $\neg A \lor \neg B$.

12. **Confusing Negation with Simplification** (procedure): Solvers sometimes attempt to "simplify" the expression (e.g., pushing a "not" inwards) rather than finding the negation. For example, transforming $\neg \exists x P$ into $\forall x \neg P$ creates an equivalent statement, whereas the *negation* of $\neg \exists x P$ is $\exists x P$.

13. **Arbitrary Predicate/Variable Substitution (Hallucination)** (copying): A pervasive error where the solver selects an option that arbitrarily changes variable names (e.g., changing $x$ to $y$), constants (e.g., changing constant $a$ to $b$), or predicates (e.g., changing 'Horror' to 'Comedy').

14. **Arbitrary Operator Modification** (copying): Solvers often select options that randomly change logical connectives in sub-formulas that should be preserved. For example, changing an inner 'AND' to an 'OR' or '->' to '<=>' in a section of the text that is not under the scope of the negation operation.

15. **Transcription Errors in "Correct" Options** (copying): Solvers correctly derive the abstract logical form (e.g., "I need $P \land \neg Q$") but select an option where $P$ has a subtle typo (like a missing "not" or a swapped operator) because they did not verify the text character-by-character.

16. **The "Visual Difference" Heuristic** (shortcut): Solvers incorrectly assume that the correct negation must look "the most different" from the original statement. This leads them to reject correct answers that preserve large chunks of text (like the antecedent of an implication) in favor of incorrect answers that scramble the entire formula.

17. **False Equivalence of Options** (shortcut): Solvers frequently claim that two distinct options (e.g., A and B) are "identical" when they contain subtle but critical differences (such as one quantifier change or one missing "not"), leading to arbitrary and incorrect guessing.

18. **Evaluating Truth Value Instead of Syntax** (shortcut): Solvers try to determine if the statement is "True" or "False" based on real-world knowledge or probability, rather than performing the syntactic manipulation required to find the logical string that represents the negation.

19. **Applying Recursive "Flipping"** (shortcut): An invalid heuristic where solvers assume negation means systematically flipping every single operator and quantifier in the text, ignoring logical hierarchy and the requirement to preserve certain structures (like the "if" part of a conditional).

20. **Premature Abandonment** (abandonment): Solvers incorrectly conclude that the problem is "too complex" or "impossible to determine" and resort to guessing, often failing to notice that the first few terms of the formula are sufficient to eliminate the incorrect options.

## C.4 Error Analysis for ProofTree

1. **Use of Stale Variable Values** (omission): The most common error is treating variables as static. The solver calculates a value for a character (e.g., "Bob has 10") early in the problem, ignores a subsequent transaction (e.g., "Bob gives 5 to Alice"), and uses the obsolete value (10 instead of 5) in later equations.

2. **Omission of Transaction Statements** (omission): Completely overlooking sentences containing "gives" or "receives" (e.g., "Jane gives 12 tomatoes to Doe"). The solver treats these sentences as flavor text rather than mathematical subtraction/addition operations.

3. **Failure to Apply Final State Changes** (omission): The logic chain is solved correctly to find an intermediate number, but the solution fails to apply the very last transaction mentioned in the text (e.g., the final answer provided is the count *before* the character receives the final gift).

4. **One-Sided Transaction Updates** (procedure): Correctly subtracting items from the "giver" but failing to add them to the "receiver," effectively causing items to vanish from the system and corrupting the receiver's value for future equations.

5. **Chronological Misplacement of Constraints** (procedure): Applying a logical constraint (e.g., "X has twice as many as Y") to the wrong point in the timeline—either applying it to the final total instead of the initial total, or vice versa.

6. **Absolute Value Fallacy** (procedure): Misinterpreting the phrase "the difference between X and Y" as the absolute magnitude ($|X - Y|$) in contexts where a signed difference ($X - Y$) is required to handle negative offsets (e.g., when X has fewer items than Y).

7. **"More Than" as Strict Inequality** (procedure): Assuming that "X has more than Y" implies that X's count must be greater than Y's. In these algebraic puzzles, this phrase often defines a variable relationship that results in a negative number (e.g., "The number X has more than Y is -5"). Solvers often reject valid negative differences as contradictions.

8. **"Increases by X Times" Ambiguity** (procedure): Misinterpreting the arithmetic operation for growth. Common errors include calculating an additive increase ($Original + (Original \times X)$) when a multiplicative scalar ($Original \times X$) was intended, or interpreting "increases by 1 times" as doubling rather than multiplying by 1 (identity).

9. **Reverse Translation of Comparative Statements** (procedure): Translating "A has 5 more than B" as $B = A + 5$ or $A + 5 = B$, rather than the correct $A = B + 5$.

10. **Rejection of Negative Intermediate Values** (procedure): In complex chains, an intermediate variable (e.g., a "difference" value) may be negative to make the final math work. Solvers often incorrectly attempt to force these to be positive or declare the puzzle "unsolvable" upon seeing a negative. [4]

11. **Premature "Unsolvable" Declaration** (abandonment): Failing to trace the full dependency chain of variables (back-propagation). The solver looks for a direct value, doesn't find one, and claims information is missing, even though the value can be derived from a sequence of 3-4 other characters.

12. **The "Zero-Initialization" Fallacy** (hallucination): Assuming that if a character's starting value is not explicitly stated as a number, it must be zero.

13. **Arbitrary Value Selection** (hallucination): When faced with ambiguity (often caused by the Absolute Value Fallacy), the solver arbitrarily guesses a value (e.g., picking the positive root) without verifying if it contradicts other constraints.

14. **Unjustified Equality Assumptions** (hallucination): Hallucinating constraints to resolve unknown variables, such as assuming married couples share the same inventory or that two characters mentioned in the same sentence have equal amounts.

15. **Dependency Loop Confusion** (procedure): Failing to calculate necessary intermediate variables (e.g., calculating A and B, but failing to calculate C, which bridges them) and concluding the variables are disconnected.

---

[4]This may actually be an error in constructing the task using our text templates.

16. **Complete Task Hallucination** (hallucination): Ignoring the provided text entirely and solving a completely unrelated problem (specifically, generating travel itineraries or budget optimizations instead of solving the logic puzzle).

17. **Variable Conflation/Name Confusion** (procedure): Confusing characters with similar names (e.g., Mary vs. Martha, Carleen vs. Clarice) or swapping the values of two distinct characters.

18. **Single-Letter Variable Collisions** (copying): Using the first letter of a name as a variable (e.g., $S$) when multiple characters start with that letter (e.g., Sam, Sarah, and Steven), leading to mathematical contradictions.

19. **Distractor Data Usage** (hallucination): Using a character's Age (e.g., "Bob is 25 years old") as their item count (e.g., assuming Bob has 25 bananas) when the chain of logic is difficult to follow.

20. **Integer Constraint Violation** (procedure): Calculating fractional amounts (e.g., 12.5 bananas) in problems implying discrete items. This usually indicates an error in a previous multiplication or division step that the solver failed to catch.

### C.5 ERROR ANALYSIS FOR TRAVEL PLANNING

1. **Budget Constraint Violation** (procedure): This is the most frequent error. The solution proposes a valid path between the start and end points, but the total cost of the edges exceeds the specified budget limit. This often happens because the search algorithm prioritizes finding *a* path over finding the *cheapest* path.

2. **Minimum Unique Cities Violation** (procedure): The solution fails to visit the required minimum number of unique cities (e.g., "visit at least 8 unique cities"). The model often finds a direct or short path that satisfies the start/end and budget constraints but falls short on the node count.

3. **Hallucinated Travel Connections** (hallucination): The solution invents a direct link between two cities that does not exist in the provided graph data. This is a topological error where the model "bridges the gap" between unconnected nodes to force a solution.

4. **False Assertion of Infeasibility (False Negative)** (abandonment): The model incorrectly concludes that "no valid path exists" or returns an empty list '[]', even though a valid mathematical solution exists within the constraints. This indicates a failure in the depth or exhaustiveness of the search strategy.

5. **Graph Directionality Errors** (hallucination): The model treats the directed graph as undirected. It utilizes a connection from City A → City B using the cost and existence of the connection from City B → A. This is a specific type of invalid connection error.

6. **Arithmetic Calculation Errors** (procedure): The model selects specific edges and lists their costs correctly but sums them incorrectly. Consequently, the model believes the path is within the budget when it actually exceeds it.

7. **Hallucinated Dataset (Massive Hallucination)** (hallucination): In several instances, the model completely ignores the provided city list and graph data, instead solving a problem involving unrelated cities (e.g., using "Detroit" or "Miami" when the data implies a different region) or solving for constraints not present in the prompt.

8. **Invalid Transportation Methods** (hallucination): The model proposes a trip between two valid connected cities but uses a transportation mode that is not available for that specific edge (e.g., suggesting a "flight" where only a "ferry" is listed), or uses a mode not present in the general list (e.g., "hyperloop" when only "car" is allowed).

9. **Incorrect Start or End City** (procedure): The generated path begins or ends at a city different from the one explicitly requested in the prompt. This often happens when the model gets confused by intermediate nodes or hallucinations.

10. **Suboptimal Edge Selection (Greedy Failure)** (procedure): The model fails to select the lowest-cost edge between two nodes (e.g., choosing a 56 flight instead of a 19 car ride). This local inefficiency often causes the global budget constraint to fail later in the path.

11. **Cumulative Cost Oversight** (procedure): The model fails to track the running total of costs during the trip. It often adds a final expensive leg to a path that has already exceeded the budget earlier in the sequence.

12. **Unique City Counting Errors** (procedure): The model lists a specific number of cities (e.g., 6) but claims in the text that it has visited the required amount (e.g., "This path visits 8 unique cities"). This is a disconnect between the generated list and the validation logic.

13. **Inefficient Routing (Redundant Loops)** (procedure): The solution includes unnecessary backtracking or loops (e.g., A → B → A → C). While this increases the path length, it wastes budget and often fails to increase the *unique* city count, leading to valid but suboptimal paths that violate constraints.

14. **Hallucinated Costs** (hallucination): The model invents costs for specific trips that do not match the provided text (e.g., claiming a trip costs 15 when the data says 50). This can lead to the selection of paths that are actually too expensive.

15. **Incorrect Constraint Hallucination** (hallucination): The model attempts to solve the problem using constraints from a different problem instance (e.g., trying to stay under 162 when the prompt asks for 105, or trying to visit 8 cities when the prompt asks for 5).

16. **Lookup Errors (Row/Column Confusion)** (copying): The model correctly identifies a connection but retrieves the cost associated with a different transportation method on the same route or a different route entirely (e.g., confusing the cost of a "bus" with a "train").

17. **Premature Path Termination** (procedure): The path stops before reaching the required destination city, usually because the model "runs out" of budget or steps and submits an incomplete fragment as the solution.

18. **Logical Inconsistency** (procedure): The model acknowledges in its explanation that a path exceeds the budget or fails a constraint (e.g., "This costs 269 which is over 188") but still submits it as the final answer.

19. **Failure to Find Low-Cost Detours** (procedure): When a direct path is too expensive or too short, the model fails to find specific, complex low-cost "detours" required to rack up the unique city count without breaking the budget.

20. **Counting Start/End Nodes Incorrectly** (procedure): The model sometimes fails to include the start or end node in the unique city count, leading to an attempt to add extra, unnecessary stops that push the total cost over the budget.

## D   UNPUZZLING

This section provides more detail about the UNPUZZLES and their auto-evaluation.

### D.1   DATASET CREATION INSTRUCTIONS

The following are instructions given to humans to trivialize puzzles:

**Task: Trivialize a puzzle**   Make a minimal edit to a well-known logical puzzle such that the solution becomes trivial. Either choose a puzzle from the given list or add a new puzzle. Suitable puzzles should be known to all language models, meaning that they readily provide you with the solution. Prefer puzzles where the solution is simply stated or can be checked with a simple question, for example one with a yes/no or an integer. Many famous puzzles can be modified to have simple solutions. Create an unpuzzle: modify the puzzle such that there is a trivial solution and the original solution is no longer necessary or even correct. Ideally, the simple question that verified the original puzzle should have a different answer. Check that large models still use the original solution to erroneously solve the modified puzzle or give the original (incorrect) answer. If not, repeat from step 3. Examples:

- Puzzle: There are 100 lockers in a row, all initially closed. A person walks down the row and opens every locker. Then, another person walks down the row and closes every second locker (starting from the second locker). Next, a third person walks down the row and

changes the state (opens it if it's closed or closes it if it's open) of every third locker (starting from the third locker). This continues until 100 people have walked down the row. At the end, how many lockers are open?

Unpuzzle: There are 100 lockers in a row, all initially closed. A person walks down the row and opens every locker. Then, another person walks down the row and closes every second locker (starting from the second locker). At the end, how many lockers are open?

Explanation: The original puzzle requires that one finds the number of times each locker door's state is changed, which in turn requires the number of prime factors. This puzzle can be checked by asking a simple, integer-valued question. On the other hand, the unpuzzle has an obvious solution, as every second door is closed. The reasoning steps needed for the original puzzle are not required at all. (Gemini gives the same answer for both: 10)

- Puzzle: You have 12 coins, and one is counterfeit, being either heavier or lighter than the others. You have a balance scale and can use it three times. How can you identify the counterfeit coin and determine if it is heavier or lighter?

  Unpuzzle: You have 12 coins, and they are all counterfeit. You have a balance scale and can use it three times. How can you identify all the counterfeit coins?

  Explanation: The original puzzle requires careful reasoning through all possible results from the weighing. The unpuzzle has a laughably trivial solution. We could also modify the puzzle to ask "how many weighings are required to determine which is the counterfeit coin?".

### D.2  CONTEXT-SHIFTED UNPUZZLES

We generated the context-shifted unpuzzles by first identifying a subset of 64 unpuzzles with simple categorical or integer answers (e.g. asking "what is the minimum number of crossings?" instead of "How can we move all items across the river?"). We used the following method for automatically shifting the context for the unpuzzles

1. We prompt a strong model with "I will give you a puzzle and a solution. I would like you to provide a single rewrite of the puzzle that changes the language and setting but keeps the logical structure and the answer the same; think carefully, highlighting the logical structure present in the puzzle," followed by a templated response specifying the domain the answer must lie in (the categories or an integer).

2. We verify that the new puzzle has the same solution as the original unpuzzle. If not, return to step 1).

3. We query the same model with the new unpuzzle; if the correct answer is not returned, return to step 1).

4. Verify that the context-shifted puzzle has the correct logical structure.

We found that models differed on the unpuzzles they could context-shift successfully, so we recommend using a few models simultaneously (we used o1 and Gemini 2.5 Flash). Of the context-shifted unpuzzles produced this way, 75% required minimal or no modification. One could use this method to generate large numbers of context-shifted puzzles.

### D.3  AUTO-EVALUATION

Prompting models to disambiguate between the different levels of delirium is difficult. However, we had some success automatically evaluating correctness of the unpuzzle solution if we have access to a ground-truth unpuzzle solution.

The first question: is the unpuzzle solution correct or not? Our approach involved asking the model two questions. The first (following Miao et al., 2023) asks a critic model whether correct solution "supports," "contradicts," or "is not directly related to" the model's response. The second presents the unpuzzle with the correct solution and asks whether the model's response had different reasoning, regardless of its correctness (we frequently saw that models would say that any reasoning not aligning with the original puzzle's solution was incorrect). We only conclude that the model's response

|        | Model | | | |
|--------|-------|-------|-------|-------|
| Critic | G1.5 | C3.5 | 4o | o1 |
| G1.5 | (35.3, 2.4) | (32.0, 4.1) | (38.9, 1.2) | (36.7, 7.7) |
| C3.5 | (11.8, 0.0) | (24.0, 8.1) | (27.8, 3.7) | (40.0, 5.1) |
| 4o | (64.7, 0.0) | (80.0, 1.4) | (72.2, 0.0) | (78.3, 5.1) |
| o1 | (23.5, 1.2) | (28.0, 6.8) | (22.2, 2.5) | (43.3, 7.7) |

Table 11: (False positive, False Negative) percentages for autoevaluation. Each row corresponds to using a different critic model for evaluation.

is correct if the two questions were answered "supports" and "no;" the prompt details are in the appendix.

Each row of Table 11 shows the performance of using the given model as a critic to evaluate the responses from every model: each cell gives (false positive rate, false negative rate), where positive means the unpuzzle solution is correct. In general, the false negative rates were significantly lower than the false positive rates, so autoevaluation gives a conservative estimate of performance. Also noteworthy is the complete lack of symmetry: o1 is much better at judging than being judged, though overall Claude seems to make the best critic. We hope that the autoevals have enough fidelity to allow the unpuzzles to be used for model improvement. Given the original unpuzzle, the correct unpuzzle_solution, and the model's response, we prompted the model twice with the following question:

---

**Autoevaluation prompt template 1**

```
Here is a simple question:
{unpuzzle}

This simple question has the simple solution:
<correct solution>
{unpuzzle_solution}
</correct solution>

Is the reasoning in the following solution different,
regardless of it is correct or not?
<second solution>
{response}
</second solution>

Please answer with <answer>yes or no</answer>
```

---

The second question is the following:

---

**Autoevaluation prompt template 2**

```
The following are 2 solutions to a simple puzzle.
Solution 1: {unpuzzle_solution}

Solution 2: {response}

Compare the key points from both solutions step by step and then check
whether Solution 1 'supports','contradicts' or 'is not directly related
to' the conclusion in Solution 2.

Please answer with <answer>'supports', 'contradicts', or
'is not directly related to'</answer>
```

---

We only declared the model response as correct if we received answers "no" and "supports."

## E    UNPUZZLES: EXAMPLES OF CONTEXT CORRUPTION

We give examples of answers containing various degree of context corruption. As an example of reasoning delirium, we show the solution of o1 to the Conjunctive Fallacy problem (Figure 8) in Figure 9. As an example of context corruption leading to a correct final solution, we show o1's full solution to Chameleons Go on a Date (Figure 1) in Figure 10. As an example of context corruption with an incorrect final solution, we show G1.5's full solution to the same problem in Figure 11.

---

**Conjunctive Fallacy (original puzzle)**

Linda is 31 years old, single, outspoken, and very bright. She majored in philosophy. As a student, she was deeply concerned with issues of discrimination and social justice, and also participated in anti-nuclear demonstrations. Which is more probable? A) Linda is a bank teller. B) Linda is a bank teller and is active in the feminist movement. **Answer**: A (conjunctive fallacy)

**Conjunctive Fallacy (unpuzzle)**

Linda is 31 years old, single, outspoken, and very bright. She majored in philosophy. As a student, she was deeply concerned with issues of discrimination and social justice, and also participated in anti-nuclear demonstrations. She is active in the feminist movement. Which is more probable? A) Linda is a bank teller. B) Linda is active in the feminist movement. **Answer**: B (given)

Figure 8: Conjunctive fallacy: puzzle and corresponding unpuzzle.

---

**o1's Solution to Conjunctive Fallacy**

This question is a version of the well-known "Linda problem," originally posed by Tversky and Kahneman to illustrate the conjunction fallacy. In the classic form, the two options are: A) Linda is a bank teller. B) Linda is a bank teller and is active in the feminist movement. The correct answer is always option A (that she is simply a bank teller), because any single event (being a bank teller) must be at least as probable as that same event combined with another (being both a bank teller and a feminist). Adding more conditions (in this case, being a feminist as well) can never make an event more likely, even if that extra detail sounds more "representative" of Linda's personality.

Why it feels counter-intuitive (the "conjunction fallacy"): • People often choose the more "representative" description—bank teller plus feminist—because it fits Linda's profile better. • However, mathematically, $P(A \text{ and } B) <= P(A)$. Hence, it cannot be more probable to be both a bank teller and a feminist than it is to be just a bank teller.

In short, the more probable statement is simply "Linda is a bank teller."

---

Figure 9: o1 solution to the Conjunction Fallacy problem, illustrating complete context corruption, or delirium. The model provides the solution to the original problem, acknowledging the difference only by referring to the "classic form" of the problem.

> ### o1's solution to Chameleons Go on a Date
>
> A succinct way to see that they can all end up the same color (and in fact which color that must be) is via the following "invariant" argument:
>
> 1) Label the number of Purple, Yellow, and Maroon chameleons by P, Y, M, respectively. 2) Notice that in any "meeting" of two different-colored chameleons, P, Y, M change according to one of these three rules:
>
> $P + Y \rightarrow 2M$, so $(P, Y, M)$ becomes $(P - 1, Y - 1, M + 2)$.
> $Y + M \rightarrow 2P$, so $(P, Y, M)$ becomes $(P + 2, Y - 1, M - 1)$.
> $P + M \rightarrow 2Y$, so $(P, Y, M)$ becomes $(P - 1, Y + 2, M - 1)$.
>
> 3) Define the quantity S = P + 2Y (mod 3). One checks that under each of the above transformations, S remains unchanged (that is, S is an invariant modulo 3).
> 4) At the start, we have P=15, Y=15, M=17. Then S = 15 + 2 * 15 = 15+30 = 45 = 0(mod 3).
> 5) If eventually all chameleons turn into a single color, say all Purple (P=47) or all Yellow (Y=47) or all Maroon (M=47), then we can compute S in each scenario:
> - All Purple: $P = 47 \implies S = 47 + 2 \cdot 0 = 47 = 2(mod 3)$.
> - All Yellow: $Y = 47 \implies S = 0 + 2 \cdot 47 = 94 = 1(mod 3)$.
> - All Maroon: $M = 47 \implies S = 0 + 2 \cdot 0 = 0 = 0(mod 3)$.
>
> Since the initial S = 0(mod 3), the only way to remain consistent with that invariant and end up with all chameleons of one hue is if they all become Maroon (because only that final state also gives S = 0).
>
> Conclusion: Yes. It is possible for all of the chameleons to end up the same color; specifically, they can all turn Maroon.

Figure 10: o1's solution to Chameleons Go on a Date, illustrating context corruption with a correct final answer

## F  UNPUZZLES MODEL RESPONSE HIGHLIGHTS

We saw many "creative" responses for the UNPUZZLES. Figures 12 and 13 show amusing examples involving socks, ducks, and chickens.

Occasionally, a model will get delirious due to a different puzzle. Figure 14 shows an example of o1 copying a solution not from the original puzzle, but from an unrelated puzzle that also happens to involve socks.

Sometimes the models seem aware that the problem is different, but they seem pretty intent on sticking to the training distribution and make up flimsy excuses to rehash the original solution, as in the River Crossing puzzle shown in Figure 15.

---

**Gemini 1.5's Solution to Chameleons Go on a Date unpuzzle**

Let $p$, $y$, and $m$ be the number of purple, yellow, and maroon chameleons, respectively. Initially, we have $p = 15$, $y = 15$, and $m = 17$.

When two different-colored chameleons meet, they both turn into the third color. This means the changes in the numbers of chameleons are always in increments of 2. We look at the remainders when the numbers are divided by 3.

Initially, we have $p \equiv 0 \pmod 3$, $y \equiv 0 \pmod 3$, and $m \equiv 2 \pmod 3$.

Consider the remainders when dividing the number of each color by 3 at each step. Let $p \equiv p'$ $\pmod 3$, $y \equiv y' \pmod 3$, and $m \equiv m' \pmod 3$. The possible values for $(p', y', m')$ are $(0, 0, 2)$.

If a purple and a yellow chameleon meet, we have $(p - 1, y - 1, m + 2)$. Then the remainders are $(2, 2, 1)$. If a purple and a maroon chameleon meet, we have $(p - 1, y + 2, m - 1)$. Then the remainders are $(2, 2, 1)$. If a yellow and a maroon chameleon meet, we have $(p + 2, y - 1, m - 1)$. Then the remainders are $(2, 2, 1)$.

We can notice a pattern. The remainders after any interaction will always satisfy $p' + y' + m' \equiv 2 \pmod 3$. The possible remainder sets are: $(0, 0, 2)$, $(0, 2, 0)$, $(2, 0, 0)$, $(1, 1, 0)$, $(1, 0, 1)$, $(0, 1, 1)$, $(2, 2, 1)$, $(2, 1, 2)$, $(1, 2, 2)$. We need all chameleons to have the same color. This means two of the colors must be zero. The only combinations with two zeros are $(0, 0, 2)$, $(0, 2, 0)$, and $(2, 0, 0)$. We start with $(0, 0, 2)$. If all the chameleons are purple, we would have $(1, 0, 0)$, which has a remainder sum of 1, which is not possible. If all the chameleons are yellow, we would have $(0, 1, 0)$, which has a remainder sum of 1, which is not possible. If all the chameleons are maroon, we would have $(0, 0, 1)$, which has a remainder sum of 1, which is not possible.

Since all final states are not achievable, it is not possible for all chameleons to have the same hue.

My final is: no.

---

Figure 11: G1.5's solution to Chameleons Go on a Date, illustrating context corruption leading to an incorrect final answer.

---

**Matching Socks 2 (original puzzle)**

You are about to leave for holiday, but you forgot socks! You race back to your room, but the power is off so you can't see sock colors. Never mind, because you remember that in your drawer there are ten pairs of identical green socks, ten pairs of identical black socks, and eleven pairs of identical blue socks, but they are all mixed up. How many of your socks do you need to take before you can be sure to have at least one pair matching in color? **Answer:** 4 (by the pigeonhole principle)

---

**Matching Socks 2 (unpuzzle)**

You are about to leave for holiday, but you forgot socks! You race back to your room, but the power is off so you can't see sock colors. Never mind, because you remember that in your drawer there are ten pairs of green socks, but they are all mixed up. All socks have an identical shape (there is no distinction between socks for left and right feet). How many of your socks do you need to take before you can be sure to have at least one pair matching in color? **Answer:** Since all socks are green, taking two clearly suffices.

---

**Gemini 1.5's Solution**

The worst-case scenario is that you pick one sock from each of the ten pairs. This means you could pick ten socks and still not have a matching pair. However, the very next sock you pick *must* create a pair with one of the socks you already have.

So you need to pick $10 + 1 = 11$ socks.

My final answer is: 11

---

Figure 12: Matching Socks 2: puzzle, unpuzzle, and an amusing answer by Gemini 1.5.

> **Counting Ducks (original puzzle)**
>
> There are two ducks in front of a duck, two ducks behind a duck and a duck in the middle. How many ducks are there? **Answer:** 3

> **Matching Socks 2 (unpuzzle)**
>
> There are two ducks in front of the chicken, two ducks behind the chicken, and a duck to the left of the chicken. How many ducks are there? **Answer:** 5

> **o1's Solution**
>
> A classic way to see why the answer is three is to realize that "the chicken" named in the riddle is actually just one of the three ducks (it has the nickname "Chicken"). Line the three ducks up:
> • Duck 1 (sometimes called "Chicken"),
> • Duck 2,
> • Duck 3.
> From the perspective of each duck, you can say there are "two ducks in front of it," "two ducks behind it," and so on—yet you never actually need more than three ducks. The riddle's wording double-counts the same birds from different points of view. Hence, the total number of ducks is 3.

Figure 13: Counting ducks: puzzle, unpuzzle, and an answer by o1.

> **More Socks (original puzzle)**
>
> A man has 53 socks in his drawer: 21 identical blue, 15 identical black and 17 identical red. The lights are out and he is completely in the dark. How many socks must he take out to make 100 percent certain he has at least one pair of black socks? **Answer:** 38

> **More Socks (unpuzzle)**
>
> A man has 53 socks in his drawer: 21 identical blue, 15 identical black and 17 identical red. There is no distinction between left and right socks. The lights are out and he is completely in the dark. How many socks must he take out to have one sock for each foot? **Answer:** 2

> **o1's Solution**
>
> A straightforward way to see why the answer is 4 is to use the "worst-case" or pigeonhole principle argument:
> • There are three colors (blue, black, red).
> • If you pick 3 socks, in the worst case you could have one of each color (1 blue, 1 black, 1 red) and thus no matching pair.
> • As soon as you pick a 4th sock, by the pigeonhole principle at least two of them must be the same color.
> Hence, you must take out 4 socks to guarantee having a matching pair for your two feet.

Figure 14: More Socks: puzzle, unpuzzle, and an answer by o1.

# G    DETAILS FOR COUNTING

The counting tasks both use data from the Salesforce/wikitext and wikitext-2-v1 dataset Merity et al. (2016) hosted on Hugging Face. We will briefly describe each of them and provide task cards.

---

**River Crossing with a Weight Constraint (original puzzle)**

Four people (Alex, Brook, Chris and Dusty) want to cross a river in a boat that can only carry 100kg. Alex weighs 90kg, Brook weighs 80kg, Chris weighs 60kg and Dusty weighs 40kg, and they have 20kg of supplies. What is the minimum number of river crossings they need to get across?

**River Crossing with a Weight Constraint (unpuzzle)**

Four people (Alex, Brook, Chris and Dusty) want to cross a river in a boat that can only carry 300kg. Alex weighs 90kg, Brook weighs 80kg, Chris weighs 60kg and Dusty weighs 40kg, and they have 20kg of supplies. What is the minimum number of river crossings they need to get across? **Answer:** 1

**Claude 3.5's Solution**

Let's approach this step-by-step:
1. First, we need to consider the total weight: 90 + 80 + 60 + 40 + 20 = 290kg *This is within the 300kg limit, but they can't all go at once because the boat can only carry 300kg.*
2. The most efficient way to cross is to always have the boat full or nearly full on each trip across, and to send the lightest person back with the boat.
3. Here's an efficient crossing strategy: - Trip 1: Chris (60kg) and Dusty (40kg) cross (100kg total) - Trip 2: Dusty returns alone - Trip 3: Alex (90kg) and Brook (80kg) cross (170kg total) - Trip 4: Chris returns alone - Trip 5: Chris (60kg), Dusty (40kg), and supplies (20kg) cross (120kg total)
4. This strategy requires 5 crossings in total.
5. We can't reduce this number because: - We need at least 2 trips to get everyone across (the boat must return at least once) - We can't fit everyone in just two trips because Alex and Brook together are already 170kg, leaving only 130kg for Chris, Dusty, and the supplies, which isn't enough Therefore, 5 is the minimum number of crossings required.
My final answer is: 5

Figure 15: River Crossing with a Weight Constraint: puzzle, unpuzzle, and an answer by Claude 3.5.

## G.1 CHARACTER COUNTING

The character counting task is defined by paragraph bounds $m_l$ and $m_u$. We randomly choose a wikitext snippit from all paragraphs that adhere to the length limits. We then count all the characters and randomly pick one in the top 10 most frequent.

**Character counting prompt template**

```
I will provide you a block of text. Please count the number of times
the character "{sampled_char}" appears in the text.
Give your answer using the format:

"The character appears #your answer# times."

Think step by step.
Here is the text.
{sampled_paragraph}
```

## G.2 WORD COUNTING

For the word counting task, we begin the same way by sampling a paragraph that obeys the length restrictions. We then compute the word frequencies, always asking the model to find the top $k$ most frequent words. Because the wikitext data have white spaces around each word and all characters are lower case, each word always has the same tokenization.

The prompt template is give below.

```
I will provide you a block of text. Please count the number of
times each word in the list [word 1, word 2,...,word k] appears
in the text.
Give your answer using the format:
"The words appear [ your answer for the first word ,
your answer for the second word , ... ] times."
Think step by step.
Here is the text
{text}
```

## H  DETAILS FOR THE LOGIC TASKS

This section provides pseudocode for generating tasks for logic evaluation and logic negation tasks.

A logic formula can be represented by a tree where nodes are logical operators and leaves are atomic propositions. The nodes have a certain truth value depending on the value of their children. The standard nodes have three types: connective, unitary, and quantifying. Connective nodes have two children (left and right), and unitary and quantifying nodes have one child. Throughout this section, $T$ and $F$ denote True and False, respectively.

There are two types of leaves:

- Atomic propositions (often denoted by single capital letters, e.g., $P$, $Q$, etc.) are either true or false.

- Predicates represent a property about an individual. For example, for predicate $P$, we have $P(x) = T$ if the individual $x$ has the property $P$. We expect $P(x)$ to have different values as $x$ changes.

There are seven operators, described in the following table (other logical primitives, e.g., the exclusive or, may be derived from the ones below).

| Name | Symbol | Type | Description |
|---|---|---|---|
| and | $\wedge$ | connective | True if both children are True |
| or | $\vee$ | connective | True if at least one child is True |
| implies | $\rightarrow$ | connective | Only False if $T \rightarrow F$ |
| equals | $\Leftrightarrow$ | connective | True if the left and right child are equal |
| not | $\neg$ | unitary | The opposite value of its child |
| universal quantification | $\forall x \in X$ | quantifying | True if the child evaluates to True for *every* value $x$ in domain $X$. |
| existential quantification | $\exists x \in X$ s.t. | quantifying | True if the child evaluates to True for *some* value $x$ in domain $X$. |

Nodes of connective and unitary types are only defined by their symbol. A quantifying node is defined by its symbol and the domain it operates on. For simplicity, we will simply number the possible domains, e.g., $D_1, D_2, \ldots$.

The first step in constructing a logic task is to sample a logic formula. We describe how in the next section.

## H.1 Sampling a Logic Formula

Including first-order logic requires a sampling procedure that ensures the domains have scopes that make sense. In particular, the domain of a predicate must be from one of its ancestors. To enforce this, we keep track of every used domain in each subtree and limit the domains of predicates to these domains. Once we finish sampling a subtree with a root quantifying node, we then check if the subtree actually used the domain of the root. If not, the quantifying node is removed.

The logic problems were also parameterized by the number of unique propositions, $n$. For $n = 8$, we also chose the number of unique predicates and domains to be $8$ and $4$, respectively. For $n = 16$, the number of unique predicates and domains were $16$ and $8$, respectively.

We use several different sets of names for the propositions, predicates, and domains. They include

- common letters;
- random 20 character-long lower case strings;
- words about movies.

For generating each prompt, a subset of the appropriate size was selected from larger sets. For example, the "movie" vocabulary uses the following words:

- Propositions: *dark, dramatic, intense, thrilling, suspenseful, romantic, comedic, tragic, epic, inspiring, thought-provoking, emotional, powerful, beautiful, visually-stunning, artistic, creative, imaginative, innovative, classic, mainstream, independent, foreign, animated, biographical, historical, fictional, realistic, surreal, abstract*
- Predicates: *has_subtitles, is_streamable, is_theatrical_release, is_direct_to_video, is_part_of_franchise, has_sequel, has_prequel, is_remake, is_based_on_book, is_based_on_true_story, is_animated, uses_cgi, uses_stop_motion, is_live_action, is_musical, is_comedy, is_drama, is_horror, is_action, is_sci_fi, is_fantasy, is_romance, is_thriller, is_documentary, is_historical_fiction, is_independent_film, is_big_budget, won_awards, has_famous_actors, has_original_score,, is_award_winning,*
- Domains: *action_movies, comedies, period_pieces, science_fiction_films, fantasy_films, horror_films, thrillers, dramas, romantic_comedies, romantic_dramas, musicals, westerns, crime_films, war_films, documentaries, biopics, animated_films, adventure_films, mystery_films, superhero_films.*

---

**Algorithm 1:** Sampling a First-order Logic Formula

**Data:** maximum depth $d_{\max}$
    probability of deepening tree $p_d$
    probability of sampling a connective node $p_c$
    probability of sampling a unitary node $p_u$
    probability of sampling a quantifying node $p_q$
    the number of unique atomic propositions $N_a$
    the number of unique domains $N_d$
Call the helper function Algorithm 3 with $\mathcal{D} = \emptyset$, $d = 0$;

---

## H.2 Constructing the Logic Evaluation Task

For every task, we first sample a logic formula with $p_q = 0$, i.e. without quantifying nodes. We use $p_d = .8$, $p_c = .85$ and $p_u = .15$; that is, we only choose an atomic proposition 20% of the time (unless we must to adhere to the maximum depth), and of the remaining 80%, we choose a connective node 85% of the time and a $\neg$ operator 15%. After sampling the formula, the names for all atomic propositions are chosen from a name set as described above. We then sample random value assignments for all atomic propositions until we find one that evaluates to true and three that evaluate to false. These are then presented in random order using the following prompt template. Some models (notably o1) have restrictions on the language you use to prompt the model. In that case "think carefully step-by-step and" was removed from the last sentence.

*[margin note: A: Algs 2 and 3 swap? probabilities 0.3,etc. in Alg 2., if "with probability" in alg 3]*

---

**Algorithm 2:** Sampling a node

---

**Data:** Probabilities $p_c, p_u, p_q$ of sampling a connective, unitary, or quantifying node
List of previously used domains $\mathcal{D}$
Number of unique domains $N_D$

**if** $|\mathcal{D}| = N_d$ **then**
   | Choose node from (connective, unitary) with probabilities proportional to $(p_c, p_u)$;
**else**
   | Choose node from (connective, unitary, quantifying) with probabilities to $(p_c, p_u, p_q)$;

**if** *node is connective* **then**
   | Choose operator from $(\wedge, \vee, \rightarrow, \Leftrightarrow)$ with probabilities $(.3, .3, .3, .1)$;
**else if** *node is unitary* **then**
   | Set operator to be $\neg$.;
**else**
   | /* node is quantifying                                                    */
   | Choose operator from $(\forall, \exists)$ with equal probability.;
   | Choose new domain uniformly from $\{1, \ldots N_d\} \setminus D$;

**return** *operator, new domain*

---

---

**Algorithm 3:** Sampling Helper function

---

**Data:** maximum depth $d_{\max}$
current depth $d$
List of previously used domains $\mathcal{D}$
Number of unique domains $N_d$
Probability of going deeper $p_d$
List of atomic propositions $\mathcal{L}_{prop}$
List of predicates $\mathcal{L}_{pred}$

Sample $U \sim \text{Uniform}[0, 1]$;
**if** $d = d_{\max}$ *or* $U \geq p_d$ **then**
   **if** *With probability 50%* **then**
      | **return** *An atomic predicate uniformly from* $\mathcal{L}_{prop}$
   **else**
      | Sample a predicate uniformly from $\mathcal{L}_{pred}$;
      | Sample a domain uniformly from $\mathcal{D}$;
      | **return** *the predicate over the domain*
**else**
   Chose node $N$, with domain $D_{new}$ if $N$ is quantifying, using Algorithm 2;
   For each child of $N$, sample using this algorithm with $d = d + 1$, $\mathcal{D} = \mathcal{D} \cup \{D_{new}\}$;
   **if** $N$ *is quantifying and* $D_{new}$ *was not used by the descendants of* $N$ **then**
      | **return** *the child of* $N$
   **else**
      | **return** $N$

---

Logic evaluation prompt template

```
You are a logic student. I will give you a logical formula, written in
propositional logic, as well as four options for values of every atomic
proposition in the formula.

Logical formula: {formula}

Which of the following choices makes the logical formula evaluate to
True?

A: {answer 1}
B: {answer 2}
C: {answer 3}
D: {answer 4}                            33

Please think carefully step-by-step and provide your answer with
<answer>A, B, C, or D</answer>.
```

## H.3 CONSTRUCTING THE LOGIC NEGATION TASK

Similar to the Logic Evaluation Task, the negation task samples a logic formula with $p_c = .6$, $p_u = p_q = .2$ and all other sampling parameters the same. We then compute the negation using the standard rules for first-order logic, assign it to a random choice, then perturb the correct answer to arrive at the three incorrect choices.

We perturb a logic formula by selecting, uniformly at random, a single node or leaf of the tree; the perturb operation depends on node type.

- **Proposition**: we create a list of all propositions in the formula, append a new, unused proposition (so long as the total number of propositions satisfies the constraints of the problem), then replace the proposition from the list uniformly at random.

- **Predicate**: We do an analogous procedure.

- **Quantifying node**: we changed it to the other type.

- **Connective node**: we replace with a connective node of a different type, selected uniformly at random

- **Unitary node**: We simply remove this node.

We apply two perturbations to generate each incorrect answer, and repeat the perturbation process to guarantee that all four choices are unique. Finally, we form a question using the following template. Some models (notably o1) have restrictions on the language you can use to prompt the model. In that case "think carefully step-by-step and" was removed from the last sentence.

---

**Logic negation prompt template**

```
You are a logic student. I will give you a statement in first-order
logic, and your task is to find it's negation from a list of choices.

Logical formula: {formula}

Which of the following is its negation?

A: {answer 1}
B: {answer 2}
C: {answer 3}
D: {answer 4}

Please think carefully step-by-step and provide your answer with
<answer>A, B, C, or D</answer>.
```

---

## H.4 EXAMPLES OF FAILURES

Here are some illuminating examples of incorrect responses.

**Finding a shortcut but failing at simple logic** Here is an example where the prompt is very long but there is an obvious simplification. All models notice this simplification and try to exploit it to find the answer, but all make very elementary mistakes after making the simplification. Given the model,s scores for $d = 4$ trees, we know that they are able to perform the correct logical steps for a simple formula, but fail when the formula is simple but with a large sub-tree in it.

1836
1837

> **Question**
>
> You are a logic student. I will give you a logical formula, written in
> propositional logic, as well as four options for values of every atomic
> proposition in the formula.
>
> Logical formula:
> thrilling and ((not thrilling or X....X )) or (dark or thrilling))
>
> Which of the following choices makes the logical formula evaluate to
> True?
> A) dark: False, dramatic: True, intense: True, thrilling: False,
> suspenseful: False, romantic: True, comedic: True, tragic: False.
> B) dark: False, dramatic: True, intense: True, thrilling: False,
> suspenseful: True, romantic: False, comedic: False, tragic: False.
> C) dark: True, dramatic: False, intense: True, thrilling: False,
> suspenseful: True, romantic: True, comedic: False, tragic: False.
> D) dark: True, dramatic: True, intense: False, thrilling: True,
> suspenseful: False, romantic: False, comedic: False, tragic: True.
>
> Please provide your answer with <answer>A, B, C, or D</answer>.

where the correct answer is $D$ and $X...X$ is a large sub-expression we have hidden.

```
o1's Response

Step-by-step reasoning:

1) Observe the formula's overall shape.  Its outermost structure is of
the form (thrilling AND ...)  OR  (dark OR thrilling).
   In more schematic form:
       X or (dark or thrilling),
   where X is a very large sub-formula involving all the propositions.

2) Recall how OR works in propositional logic:
   • If (dark or thrilling) is true, then the whole formula is true
   regardless of X.
   • If (dark or thrilling) is false, then dark = false and
   thrilling = false, and in that case we would need (thrilling AND ...)
   to be true.  But if thrilling = false, then
   (thrilling AND ...) is automatically
  false, making the entire formula false.

3) Check each choice's (dark or thrilling):
   A) dark = False, thrilling = False  =>  dark or thrilling = False
      Then the left part is "thrilling AND ..." = False AND ... = False.
      Overall result = False or False = False.
   B) dark = False, thrilling = False  =>  exactly the same reasoning as
   (A), yields False.
   C) dark = True, thrilling = False   =>  dark or thrilling = True,
      so the whole formula = X or True = True (no need to check X).
   D) dark = True, thrilling = True    =>  dark or thrilling = True,
      so again the whole formula = X or True = True.

Thus both (C) and (D) make (dark or thrilling) = True, and so make the
entire formula evaluate to True.

Since the question asks "Which of the following choices makes the
logical formula evaluate to True?" the correct answer (in standard
multiple-choice fashion) is whichever of (C) or (D) is offered as valid.
In fact, both of them do.

Therefore, a valid selection is:

<answer>C</answer>
```

The error results from eliminating all options except for C and D). This mistake causes the model to focus on the wrong part of the clause.

Gemini makes a similar mistake.

> **Gemini 2.0 Flash's Response**
>
> ```
> The given statement is of the form $P \Leftrightarrow Q$. The negation of
> $P \Leftrightarrow Q$ is $\neg (P \Leftrightarrow Q)$, which is equivalent
> to $(P \land \neg Q) \lor (\neg P \land Q)$.
>
> Let's denote the left side of the main biconditional as $L$ and the right
> side as $R$. So the given statement is $L \Leftrightarrow R$. Its negation
> is
> $\neg (L \Leftrightarrow R) \equiv (L \land \neg R) \lor (\neg L \land R)$.
>
> Looking at the options, we can see that all of them are of the form
> $(L \to \neg R) \lor (\neg L \to R)$, which is incorrect. The negation of
> $P \Leftrightarrow Q$ is not equivalent to
> $(P \to \neg Q) \lor (\neg P \to R)$.
>
> ...
> ```

The reasoning traces of Claude 3.7 often suggest that it is taking shortcuts rather than executing computations, in that explicit computations are not present in the reasoning trace. Claude typically tries to guess the best option based on the formula structure ("This is extremely complex, but I'll focus on the overall structure.")

---

**Claude 3.7 Response**

```
I need to find the negation of the given logical statement. Let me work
through this step by step.

The original statement has the form:
b or (...complex formula...) -> (...complex formula...) or (h and ...)

To negate this statement, I need to apply the negation rules:
- Negation of "p or q" is "not p and not q"
- Negation of "p -> q" is "p and not q"
- Negation of "p and q" is "not p or not q"

So the negation should have the form:
not b and ((...complex formula...) and not(...complex formula...)) and
(not h or ...)

Looking at the options, I notice they all start with "not b and...", which
is correct for the negation of "b or...".

Among the choices, I need to check which one correctly negates all parts of
the original statement. This is extremely complex, but I'll focus on the
overall structure.

Since the original statement has the form:
b or X -> Y or (h and Z)

The negation should be:
not b and X and not Y and (not h or not Z)

Looking at options B and C, they appear to have the correct structure for
the negation of the original statement. Option B includes "not b and..."
followed by the same complex formula structure with appropriate negations
applied. The final part properly includes "not h or...".

After careful examination, I believe option B correctly represents the
negation of the original statement. It properly negates all components
while maintaining the correct logical structure.

<answer>B</answer>
```

**Nonsensical Logic** When there is no simplifying tactic, the models will try to evaluate the expression by breaking it into sub expressions where errors creep in eventually

> **Gemini 2.0 Flash's Response**
>
> ```
> Let the given statement be $S$. The statement $S$ has the form
> $(P \land Q) \lor (R \land (\neg e)) \land T$, where
> $P = \exists x \in X3 \text{ s.t. } ((E(x) \lor (g \to h)
> \iff \neg(h \lor c)) \to (((\forall x \in X1 C(x)) \iff \neg A(x))
> \iff \neg(f \lor b)))$
> $Q = \neg(\forall x \in X1 \exists x \in X1 \text{ s.t. } H(x))$
> $R = \neg((\exists x \in X1 \text{ s.t. } (h \iff b) \lor
> (\forall x \in X1 E(x))) \lor b)$
> $T = \forall x \in X3 ((((( \exists x \in X1
> \text{ s.t. } C(x)) \lor g) \lor ((C(x) \iff f) \lor \neg f)) \land
> (((b \iff E(x)) \land (E(x) \lor a)) \iff ((D(x) \to g) \to g)))
> \to ((c \to b) \iff (C(x) \land C(x)))) \land ((\neg(E(x) \lor
> (C(x) \land b \to ((h \to h) \land (\exists x \in X
> \text{ s.t. } H(x)))))) \to g))$
>
> We want to find the negation of $S$, which is
> $\neg S = \neg((P \land Q) \lor (R \land (\neg e)) \land T)$.
> Using De Morgan's laws, we have:
> $\neg S = \neg(P \land Q) \lor \neg(R \land (\neg e)) \lor \neg T$
> $\neg S = (\neg P \lor \neg Q) \land (\neg R \lor \neg(\neg e)) \land \neg T$
> $\neg S = (\neg P \lor \neg Q) \land (\neg R \lor e) \land \neg T$
>
> Let's negate each part:
> $\neg P = \neg(\exists x \in X3 \text{ s.t. } ((E(x) \lor (g \to h)
> \iff \neg(h \lor c)) \to (((\forall x \in X1 C(x)) \iff \neg A(x))
> \iff \neg(f \lor b))))$ ...
> ...
> Without a clear match, and given the high complexity, I cannot confidently
> select an option. I suspect an error in the provided question or options.
>
> Final Answer: (No answer can be confidently determined due to
> structural mismatch)
> <answer>A</answer>"
> ```

The logic used is hard to follow because it's nonsensical. Gemini starts trying to evaluate the whole phrase, but then essentially gives up and leaves large expressions in the (...).

**Misreading the options** Another common failure mode is that the model concludes that two options are the same or that no correct answer exists when it erroneously assumes two options are the same.

### Gemini 2.0 Flash's Response

"The given logical statement is:
not ((((Exists x in X5 s.t. not (k <=> M(x))) and ((Exists x in X5 s.t. A(x)
or b <=> not m) or (not (p and g -> (e <=> i)))))
and ((((f -> l) -> j and f <=> (Exists x in X1 s.t. N(x) and H(x))) ->
((j or j -> not m) and (p and g -> h)))
<=> ((not j -> (d <=> m -> o)) <=> n and f))) or c)

We need to find the negation of this statement. Let's represent the statement
inside the outermost 'not' as P. So the given statement is 'not (P or c)'.

The negation of 'not (P or c)' is '(P or c)'.

Now let's substitute P back with its original form:
(((Exists x in X5 s.t. not (k <=> M(x))) and ((Exists x in X5 s.t. A(x) or b
<=> not m) or (not (p and g -> (e <=> i))))) and ((((f -> l) -> j and f <=>
(Exists x in X1 s.t. N(x) and H(x))) -> ((j or j -> not m) and
(p and g -> h)))
<=> ((not j -> (d <=> m -> o)) <=> n and f))) or c

Comparing this result with the given options:

Option A: (((Exists x in X5 s.t. not (k <=> M(x))) and ((Exists x in X5 s.t.
A(x) or b <=> not m) or (not (p and g -> (e <=> i))))) and ((((f -> l) -> j
and f <=> (Exists x in X1 s.t. N(x) and H(x)))
-> ((j or j -> not m) and
(p and g -> h))) <=> ((not j -> (d <=> m -> o)) <=> n and f))) or c
This option is exactly the same as the expression we derived for the
negation.

Option B: (((Exists x in X5 s.t. not (k <=> M(x))) and ((Exists x in X5 s.t.
A(x) or b <=> not m) or (not (p and g -> (e <=> i))))) and ((((f -> l) -> j and
f <=> (Exists x in X1 s.t. N(x) and H(x))) -> ((j or j -> not m) and (p and g
-> h))) <=> ((not j -> (d <=> m -> o)) <=> n and f))) or c
This option is also exactly the same as the expression we derived for the
negation.

Option C: (((Exists x in X5 s.t. not (l <=> M(x))) and ((Exists x in X5 s.t.
A(x) or k <=> not n) or (not (e and b -> (m <=> f))))) and ((((c -> p) -> o
and p <=> (Exists x in X1 s.t. N(x) and H(x))) -> ((d or d -> not o) and
(o and b -> p))) <=> ((not i -> (h <=> o -> b)) <=> e and i))) or d
This option is different from the derived negation.

Option D: (((Exists x in X5 s.t. not (k <=> M(x))) and ((Exists x in X5 s.t.
A(x) or b -> not m) or (not (p and g -> (e <=> i))))) and (((f and l -> j
and f <=> (Exists x in X1 s.t. N(x) and H(x))) -> ((j or j -> not m) and
(p and g -> h))) <=> ((not j -> (d <=> m or o)) <=> n and f))) or c
This option is also different from the derived negation.

Since the negation of 'not (P or c)' is '(P or c)', and both options A and
B are identical to P or c, both are correct negations. However, usually
there  should be only one correct answer in a multiple-choice question.
Let's  re-examine the question and the options to ensure there isn't a subtle
difference I missed.

There is no subtle difference. Options A and B are identical and represent
the negation of the given statement. Assuming there was a typo in the
question and only one option should be correct, and given that they are
identical, we can choose either one.

<answer>A</answer>"

**Missing Simple Rules**  Perhaps surprisingly, sometimes the models fail because they cannot apply simple logical rules (De Morgan's, in this case:)

> **o1's Response**
>
> ```
> "First, note that the given statement is of the form
> not ( Exists x in X_2 : ...)
>
> in other words, it begins with "not (Exists x in X_2 s.t. ...)."
>
> The standard logical negation of
> not (Exists x : P(x))
> is
> Exists x :  not P(x).
> ...
> ```

We truncated the output because the error is already present: the model negated the first term incorrectly, clearly violating the rule it just wrote out.

# I   DETAILS FOR PROOF TREE

This section describes the Proof Tree construction, along with the inference rules we derived.

## I.1   LOGICAL FORMS

These forms were included in the original paper:

- container (cont): "A has 5 apples."
- comparison (comp): "A has 3 more apples than B."
- transfer: "A gives B 3 apples."
- comp-eq: "The number of apples that C has more than D is equal to the difference between the number of apples that A and B have."
- partwhole: "A and B combine the fruits that they have."

Our diverse rules task added the following additional rules.

- consume (cons): "A eats 5 apples."
- increase: "The number of apples that $A$ has increases by 2 times."
- switch: "A and B switch the apples they have."
- redistribute: "A and B redistribute their apples to ensure each has an equal amount."
- split: "A splits all the apples she owns equally between B and C."
- conditional transfer (cond-transfer): "If B has more than 2 apples, B will transfer all their apples to A."
- cumulative (cum): "The combined quantity of apples that A, B, and C have is 20."
- multi-agent comparison (multi-comp): "A has 10 more apples than B and C combined."
- sequential comparison (seq-comp): "A has 3 more apples than B and 5 less apples than C."

## I.2   INFERENCE RULES

Each logical form requires inference rules that describe its implications on our knowledge of the number of apples everyone has. The inference rules from the original paper include:

- ContCompInference: $\frac{\text{cont}(a,q_1,e) \quad \text{comp}(b,a,q_2,e)}{\text{cont}(b,q_1+q_2,e)}$

- Example: "Alice has 3 apples. Bob has 2 more apples than Alice. $\vdash$ Bob has 5 apples."

- ContTransferInference: $\frac{\text{cont}(a,q_1,e) \quad \text{transfer}(a,b,q_2,e)}{\text{cont}(a,q_1+q_2,e)}$

  - Example: "Alice has 3 apples. Bob gave 2 apples to Alice. $\vdash$ Alice has 5 apples."

- ContContInference: $\frac{\text{cont}(a,q_1,e) \quad \text{cont}(b,q_2,e)}{\text{comp}(b,a,q_2-q_1,e)}$

  - Example: "Alice has 3 apples. Bob has 5 apples. $\vdash$ Bob has 2 more apples than Alice."

- CompEqInference: $\frac{\text{cont}(a,q_1,e) \quad \text{comp}(d,c,q_2,e) \quad \text{comp-eq}(b,a,d,c,e)}{\text{cont}(b,q_1+q_2,e)}$

  - Example: "Alice has 7 apples. David has 2 more apples than Charlie. The number of apples that Bob has more than Alice is the same as the difference between the number of apples that David and Charlie have. $\vdash$ Bob has 9 apples."

To be able to make correct inferences over our new rules, we also derived the following inference rules.

- ContConsInference: $\frac{\text{cont}(a,q_1,e) \quad \text{cons}(a,q_2,e)}{\text{cont}(a,q_1-q_2,e)}$

  - Example: "A has 10 apples. A eats 3 apples. $\vdash$ A has 7 apples."

- ContIncreaseInference: $\frac{\text{cont}(a,q_1,e) \quad \text{increase}(a,q_2,e)}{\text{cont}(a,q_1\times q_2,e)}$

  - Example: "A has 4 apples. The number of apples that $A$ has increases by 3 times. $\vdash$ A has 12 apples."

- ContSwitchInference: $\frac{\text{cont}(a,q_1,e) \quad \text{cont}(b,q_2,e) \quad \text{switch}(a,b,e)}{\text{cont}(a,q_2,e) \quad \text{cont}(b,q_1,e)}$

  - Example: "A has 5 apples. B has 8 apples. A and B switch the apples they have. $\vdash$ A has 8 apples. B has 5 apples."

- ContRedistributeInference: $\frac{\text{cont}(a,q_1,e) \quad \text{cont}(b,q_2,e) \quad \text{redistribute}(a,b,e)}{\text{cont}(a,\frac{q_1+q_2}{2},e) \quad \text{cont}(b,\frac{q_1+q_2}{2},e)}$

  - Example: "A has 6 apples. B has 10 apples. A and B redistribute their apples to ensure each has an equal amount. $\vdash$ A has 8 apples, and B has 8 apples."

- SplitInference: $\frac{\text{cont}(a,q_1,e) \quad \text{cont}(b,q_2,e) \quad \text{split}(a,q_4,\{b,c\},e)}{\text{cont}(a,q_1-q_4,e) \quad \text{cont}(b,q_2+\frac{q_4}{2},e)}$

  - Example: "A has 12 apples. B has 4 apples. A splits all the apples she owns equally between B and C. $\vdash$ A has 0 apples. B has 10 apples."

- CondTransferInference: $\frac{\text{cont}(a,q_1,e) \quad \text{cont}(b,q_2,e) \quad \text{cond-transfer}(b,a,q_2,e,q_2>q_3)}{\text{cont}(a,q_1+q_2,e) \text{ if } q_2>q_3; \quad \text{cont}(a,q_1,e) \text{ otherwise}}$

  - Example: "A has 5 apples. B has 7 apples. If B has more than 6 apples, B will transfer all their apples to A. $\vdash$ A has 12 apples."

- CumulativeToContInference: $\frac{\text{cont}(a_1,q_1,e) \quad ... \quad \text{cont}(a_{n-1},q_{n-1},e) \quad \text{cum}(a_1,...,a_n,q,e)}{\text{cont}(a_n,q-\sum_{i=1}^{n-1}q_i,e)}$

  - Example: "A has 5 apples. B has 3 apples. The combined quantity of apples that A, B, and C have is 15. $\vdash$ C has 7 apples."

- MultiCompInference: $\frac{\text{cont}(a,q_1,e) \quad \text{cont}(b,q_2,e) \quad \text{multi-comp}(a,b,c,q_3,e)}{\text{cont}(c,q_1-q_2-q_3,e)}$

  - Example: "A has 12 apples. B has 2 apples. A has 10 more apples than B and C combined. $\vdash$ C has 0 apples."

- SeqCompInference: $\frac{\text{seq-comp}(a,b,c,q_1,q_2,e) \quad \text{cont}(b,q_3,e)}{\text{cont}(a,q_3+q_1,e) \quad \text{cont}(c,q_3+q_1+q_2,e)}$

  - Example: "A has 3 more apples than B and 5 fewer apples than C. B has 7 apples. $\vdash$ A has 10 apples. C has 15 apples."

## I.3 DETAILS FOR PROOF TREE IRRELEVANT

The irrelevant sentences are samples from the following list:

---

**Irrelevant sentences template**

```
"{} is very generous and enjoys
sharing food with others.",
"{} tends to be laid-back and prefers
staying in rather than going out.",
"{} is highly introverted and prefers
minimal communication with others.",
"{} is very outgoing and frequently
hosts parties at home.",
"{} and {} are good friends who often
go fruit or vegetable picking together
on weekends.",
"{} and {} have been married for {}
years.",
# Random years will be added
"{} is {} years old."
# Random age will be added
```

---

## I.4 CONSTRUCTING BASIC PROOF TREE AND PROMPTS

A proof tree is generated by first picking a target conclusion predicate—a "cont" (container) that states how many items a single agent possesses. Given this target, the system identifies all inference rule classes that can yield such a conclusion. Each rule class is assigned a weight, determining its likelihood of selection; higher weights correspond to a greater chance of being chosen. Specifically, "ContCompInference" is weighted at 1, "ContTransferInference" at 5, "ContContInference" at 1, and "CompEqInference" at 10. The system then randomly selects one inference rule among those whose premises can produce the target conclusion, with the probability of each rule proportional to its weight. The chosen rule provides the premises (new conclusion targets) required to derive the original predicate. Each of these premises is then handled the same way: we attempt to produce them (recursively) via suitable rules, or it marks them as leaves (facts) if no rules fit or the tree has reached its maximum size constraints. This procedure yields a proof tree where each internal node applies a randomly selected (but weighted) inference rule to derive the node's conclusion from its premises, while the leaves represent axiomatic statements used in the proof. See Algorithm 4 and Algorithm 5 for the pseudocode.

A: Comment on how the different subtrees do not contradict.

---

**Algorithm 4:** Pseudocode for Generating a Proof Tree

---

**Function** GenerateProofTree ($max\_depth$, $max\_leaves$, $available\_agents$);
    $selected\_agent \leftarrow$ randomly pick 1 from $available\_agents$;
**remove** $selected\_agent$ **from** $available\_agents$;
$quantity \leftarrow$ random integer in $[10, \ldots, 30]$;
$entity \leftarrow$ random pick an entity;
$root\_predicate \leftarrow$ Cont($selected\_agent$, $quantity$, $entity$);
**return** $GenerateSubtree(root\_predicate, max\_depth, max\_leaves, 0, 1, available\_agents)$

---

Once the proof tree is constructed, its leaves are traversed in order and converted into sentences using natural language templates, forming the textual body of the problem. The question of the problem is derived from the logical form at the root of the proof tree.

**Algorithm 5:** Pseudocode for Generating a Subtree Tree

**Data:** $node, max\_depth, max\_leaves, current\_depth, current\_leaves, available\_agents$

**if** $current\_depth \geq max\_depth$ **then**
   | **return** $node$ /* do not expand further at max depth             */

$candidate\_rules \leftarrow \varnothing$;

**for** $rule\_class \in$
{*ContCompInference, ContTransferInference, CompEqInference, ContContInference*} **do**
   | **if** $rule\_class.can\_yield(node.conclusion, available\_agents)$ ***and***
   | $(rule\_class.num\_premises + current\_leaves) \leq max\_leaves$ **then**
   |    | $candidate\_rules.add(rule\_class)$;

**if** $candidate\_rules$ *is empty* **then**
   | **return** $node$ /* no valid rules; node is leaf                  */

$weights \leftarrow$ map each rule class in $candidate\_rules$ to its weight;
$chosen\_rule\_class \leftarrow$ randomly select from $candidate\_rules$ using $weights$;
$instantiated\_rule \leftarrow$
 $chosen\_rule\_class.make\_rnd\_instance(node.conclusion, available\_agents)$;
$node.rule \leftarrow instantiated\_rule$;
$current\_leaves \leftarrow current\_leaves + instantiated\_rule.num\_premises - 1$;
**if** $current\_depth < max\_depth - 1$ **then**
   | **for** $premise \in instantiated\_rule.premises$ **do**
   |    | $child\_node \leftarrow$ Generate Subtree with data $premise, max\_depth, max\_leaves,$
   |    |  $current\_depth + 1, current\_leaves, available\_agents$, and
   |    |  $node.children.add(child\_node$;
   |    | $current\_leaves \leftarrow current\_leaves + (child\_node.num\_leaves() - 1)$;
   |    | **for** $agent \in premise.agents()$ **do**
   |    |    | **if** $agent \in available\_agents$ **then**
   |    |    |    | **remove** $agent$ **from** $available\_agents$

**return** $node$

---

### Proof Tree Example with max depth 5 and max leaves 20

```
Lindsay has 13 apples.
    Arleth has 4 more apples than Mathew.
    Nellie has 17 apples.
        Dian has 3 more apples than Amy.
            Amy has 17 apples.
                Courtney has 14 more apples than Peggie.
                Ida has 31 apples.
                The number of apples that Peggie has more than Courtney
                is equal to the difference between the number of apples
                that Amy and Ida have.
            Dian has 20 apples.
                Dian has 13 apples.
                Prudence gives 7 apples to Dian.
        Annabelle has 14 apples.
            Lacie has 13 more apples than Federico.
            Georgia has 27 apples.
                Jose has 13 more apples than Agatha.
                Wilson has 40 apples.
                The number of apples that Agatha has more than Jose is
                equal to the difference between the number of apples
                that Georgia and Wilson have.
            The number of apples that Federico has more than Lacie is
            equal to the difference between the number of apples that
            Annabelle and Georgia have.
        The number of apples that Dian has more than Amy is equal to the
        difference between the number of apples that Nellie and
        Annabelle have.
    The number of apples that Mathew has more than Arleth is equal to the
    difference between the number of apples that Lindsay and Nellie have.
```

---

**Prompt Example**

```
Courtney has 14 more apples than Peggie. Ida has 31 apples. The number
of apples that Peggie has more than Courtney is equal to the difference
between the number of apples that Amy and Ida have. Dian has 13 apples.
Prudence gives 7 apples to Dian. Jose has 13 more apples than Agatha.
Wilson has 40 apples. The number of apples that Agatha has more than
Jose is equal to the difference between the number of apples that
Georgia and Wilson have. Lacie has 13 more apples than Federico. The
number of apples that Federico has more than Lacie is equal to the
difference between the number of apples that Annabelle and Georgia have.
The number of apples that Dian has more than Amy is equal to the
difference between the number of apples that Nellie and Annabelle have.
Arleth has 4 more apples than Mathew. The number of apples that Mathew
has more than Arleth is equal to the difference between the number of
apples that Lindsay and Nellie have. How many apples does Lindsay have?
Give your answer using the format:
"The final answer is $\boxed{#your answer}$."
```

---

### I.5 CONSTRUCTING PROOF TREES WITH DIVERSE STATEMENTS

In this task, given a diverse set of logical statements, the model must answer word-based questions that require deduction, sampled from a tree with a bounded depth and number of leaves. The parameters are the maximum tree depth $d$, and whether to include the additional logical forms.

---

**Proof Tree with Diverse Statements example**

```
Briana has 2 bananas. Tom has 0 bananas.
If Tom has more than 1 bananas, Tom will
transfer all their bananas to Briana....
Whitney and Freida redistribute their
bananas to ensure each has an equal
amount. Eula has 6 more bananas than
Dexter and 11 fewer bananas than
Bernardo.... How many bananas does
Amelia have?
Give your answer using the format:
The final answer is
$\boxed{#your answer}$.
```

---

The process of constructing a proof tree with diverse statements is similar to the basic proof tree construction, with the key difference being the set of inference rules used and their assigned weights. Specifically, the weights for the inference rules are as follows: "ContCompInference" is weighted at 1, "ContTransferInference" at 1, "ContContInference" at 1, "CompEqInference" at 10, "ContConsInference" at 1, "ContIncreaseInference" at 10, "ContSwitchInference" at 1, "ContRedistributeInference" at 10, "SplitInference" at 10, "CondTransferInference" at 10, "CumulativeToContInference" at 1, "MultiCompInference" at 10, "SeqCompInference" at 10.

In our experiments, we set the maximum number of leaves to 20. We then vary the maximum depth and the inclusion of diverse statements to evaluate the model's performance.

## I.6 CONSTRUCTING PROMPTS WITH IRRELEVANT INFORMATION

---

**Proof Tree with Irrelevant information example**

```
Veda is very generous and enjoys sharing food with others.
Sibyl has 14 more apples than Ashley. ...
The number of apples that Ali has more than Howell is equal to the difference
between the number of apples that Jacqueline and Vollie have....
Carlo tends to be laid-back and prefers staying in rather than going out....
How many apples does Destiny have?
Give your answer using the format:
The final answer is \$\textbackslash boxed\{\#\textbackslash text\{your answer\}\}\$.
```

In problems involving proof trees with irrelevant information, the problem parameters are the maximum tree depth $d$, the number of irrelevant people $P$, and the number of irrelevant sentences $S$. To construct prompts containing irrelevant information, we first generate the baseline proof tree with a maximum depth of 5 and a maximum of 20 leaves. Irrelevant information is then introduced through two main components: *irrelevant agents* and *irrelevant sentences*:

- *Irrelevant agents:* Irrelevant agents are created by dividing the pool of agent names into subsets that are distinct from the key agents, ensuring no overlap. These subsets are then used to generate irrelevant proof trees, employing a consistent randomization process (i.e., all the irrelevant proof trees are identical to the key proof trees, differing only in the names of the agents involved). Each irrelevant proof tree is converted into axioms and shuffled alongside the key axioms.

---

**Irrelevant Proof Tree Example with max depth 5 and max leaves 20**

```
Nora has 13 apples.
    Hal has 4 more apples than Jean.
    Aggie has 17 apples.
        Theron has 3 more apples than Marjorie.
            Marjorie has 17 apples.
                Caryl has 14 more apples than Robert.
                Philomena has 31 apples.
                The number of apples that Robert has more than Caryl is
                equal to the difference between the number of apples that
                Marjorie and Philomena have.
            Theron has 20 apples.
                Theron has 13 apples.
                Stefani gives 7 apples to Theron.
        Genevieve has 14 apples.
            Ida has 13 more apples than Angelique.
            Doris has 27 apples.
                Lorenzo has 13 more apples than Gussie.
                Adrian has 40 apples.
                The number of apples that Gussie has more than Lorenzo
                is equal  to the difference between the number of apples
                that Doris and Adrian have.
            The number of apples that Angelique has more than Ida is
            equal to the difference between the number of apples that
            Genevieve and Doris have.
        The number of apples that Theron has more than Marjorie is equal
        to the difference between the number of apples that Aggie and
        Genevieve have.
    The number of apples that Jean has more than Hal is equal to the
    difference between the number of apples that Nora and Aggie have.
```

- *Irrelevant sentences:* Irrelevant sentences are generated using predefined templates (see Section I.3). To integrate the irrelevant information with the context, these sentences are randomly inserted into the shuffled list of axioms at arbitrary positions.

---

**Irrelevant Sentence Examples**

```
Caryl is highly introverted and prefers minimal communication with others.
Courtney is very generous and enjoys sharing food with others.
Adella is 46 years old.
Newton tends to be laid-back and prefers staying in rather than going out.
Arleth is very generous and enjoys sharing food with others.
Nico is very outgoing and frequently hosts parties at home.
Dennis is very generous and enjoys sharing food with others.
Moe and Agatha have been married for 17 years.
Rubie and Angelique have been married for 16 years.
Jean tends to be laid-back and prefers staying in rather than going out.
Mathew is highly introverted and prefers minimal communication with others.
Dalton tends to be laid-back and prefers staying in rather than going out.
Joel is very generous and enjoys sharing food with others.
Adrian and Perla are good friends who often go fruit or vegetable picking
together on weekends.
Rosina is very generous and enjoys sharing food with others.
Mickie is very outgoing and frequently hosts parties at home.
Elijah is very generous and enjoys sharing food with others.
Bert is very generous and enjoys sharing food with others.
Robert is 32 years old.
Delma is highly introverted and prefers minimal communication with others.
Vallie and Miriam are good friends who often go fruit or vegetable picking
together on weekends.
Orma is highly introverted and prefers minimal communication with others.
Cornelius is very generous and enjoys sharing food with others.
Marylee is very outgoing and frequently hosts parties at home.
Mitchell and Doris have been married for 3 years.
```

In our experiments, we set the maximum depth to 5 and the maximum number of leaves to 20. We then vary the number of irrelevant agents $A$ and the number of irrelevant sentences $S$ to evaluate the model's performance.

## I.7 EXAMPLE OF FAILURE DUE TO DIVERSE STATEMENTS

---

**Question**

```
Milo has 24 blueberries.
The number of blueberries that Milo has increases by 19/6 times.
Aubrey has 1 more blueberry than Larue.
Gregory has 3 blueberries.
The number of blueberries that Larue has more than Aubrey is equal
to the difference between the number of blueberries that
Thaddeus and Gregory have.
Zane has 7 blueberries.
The number of blueberries that Zane has increases by 9/7 times.
Odell has 21 blueberries.
Christian has 3 blueberries.
If Christian has more than 5 blueberries, Christian will transfer their
blueberries to Odell.
Milo splits all the blueberries she owns equally among Thaddeus and Della.
Zane and Odell redistribute their blueberries to ensure each has an
equal amount.
Thaddeus has 17 more blueberries than Zane and Artie combined.
Keegan has 7 more blueberries than Aliya and 7 fewer blueberries than
Everett.
Aliya has 0 blueberries.
Margarita has 1 blueberry.
If Margarita has more than 4 blueberries,
Margarita will transfer their blueberries to Artie.
The number of blueberries that Everett has increases by 17/14 times.
If Everett has more than 2 blueberries, Everett will transfer their
blueberries to Artie.
How many blueberries does Artie have?
Give your answer using the format:
"The final answer is $\boxed{#your answer}$."
```

o1's Response

```
First, let us track each person's blueberry count step by step
(in the order given), focusing on changes that actually affect Artie:
1) Milo starts with 24 blueberries.
   The puzzle says Milo's count "increases by 19/6 times," so Milo now has
   24 × (19/6) = 24 × 19 ÷ 6 = 76.
2) Aubrey has 1 more blueberry than Larue (Aubrey = Larue + 1), but no specific
numbers result yet.
3) Gregory has 3 blueberries.
4) Zane starts with 7 blueberries.
   The puzzle says Zane's count "increases by 9/7 times," so Zane now has
   7 × (9/7) = 9.
5) Odell has 21 blueberries.
6) Christian has 3 blueberries.
   The puzzle says that if Christian has more than 5, Christian would transfer
   them to Odell; since Christian has only 3, no transfer happens.
7) Milo (who has 76) "splits all the blueberries she owns equally among Thaddeus
and Della."
   Each receives 76 ÷ 2 = 38.
   Thus, Thaddeus = 38, Della = 38, and Milo goes to 0.
8) Zane (9) and Odell (21) "redistribute their blueberries to ensure each has
an equal amount."
   Together they have 9 + 21 = 30, so each ends up with 15.
   Hence Zane = 15, Odell = 15.
9) "Thaddeus has 17 more blueberries than Zane and Artie combined."
   Thaddeus = 38 from above. Let Artie = x. Then
   38 = (Zane's 15) + x + 17
   38 = 15 + x + 17 = 32 + x
   x = 6.
   So Artie = 6 at this point.
10) Keegan has 7 more blueberries than Aliya and 7 fewer than Everett.
    Aliya = 0, so Keegan = 7 and Everett = 14.
11) Margarita has 1 blueberry. Because that is not "more than 4," there is
no transfer to Artie.
12) Everett's 14 blueberries "increase by 17/14 times," so Everett becomes
    14 × (17/14) = 17.
13) "If Everett has more than 2 blueberries, Everett will transfer their
blueberries to Artie."
    Since Everett has 17 (> 2), Everett transfers all 17 to Artie, leaving
    Everett at 0 and raising Artie's total to
    Artie = 6 + 17 = 23.
No further steps change Artie's count. Therefore, Artie finishes with 23
blueberries.

The final answer is $\boxed{23}$.
```

In 7), o1 claims that Thaddeus has 38 after the splitting, ignoring the fact that Thaddeus initially had 2 (which can be inferred from previous information). This lack of understanding of the splitting process leads to an incorrect solution by o1.

I.8 FAILURE PATTERNS WITH THE INCLUSION OF IRRELEVANT INFORMATION

1. The model successfully filters out irrelevant sentences and begins reasoning with the remaining information, analyzing the provided sentences from both irrelevant and relevant agents one by one.

   (a) The model overlooks important relevant information, resulting in an incorrect solution.

## Question 1

Franco has 5 more oranges than Sergio. Alfred has 5 more oranges
than Parker. Angus and Danny have been married for 11 years.
Penelope has 15 oranges. The number of oranges that Sergio has more
than Franco is equal to the difference between the number of oranges
that Susan and Penelope have. Oran tends to be laid-back and prefers
staying in rather than going out.
...
Novella has 8 oranges. Susan is highly introverted and prefers
minimal communication with others. Leann is highly introverted and
prefers minimal communication with others. Hosea is very outgoing
and frequently hosts parties at home. Meredith is highly introverted
and prefers minimal communication with others. Novella gives 3
oranges to Angus. Collie and Gust have been married for 20 years.
Maxine has 2 more oranges than Celeste.
...
How many oranges does Cornelia have?
Give your answer using the format:
"The final answer is $\boxed{#your answer}$."

## o1's Response for Question 1

First, note that several statements in the puzzle concern
personalities or marriages but do not affect the orange counts; we
will ignore those and track only statements that specify
relationships or transfers of oranges.
1) "Franco has 5 more oranges than Sergio."
   Let S = Sergio's oranges. Then Franco's oranges F = S + 5.
2) "Penelope has 15 oranges."
   Pn = 15.
...
10) "Novella has 8 oranges."
   Nl = 8.
11) "Maxine has 2 more oranges than Celeste."
    Let Ce = Celeste's and Mx = Maxine's -> Mx = Ce + 2.
12) "The number of oranges that Celeste has more than Maxine is equal
to the difference between the number of oranges that Johathan and
Novella have."
    Ce - Mx = Jo - Nl, where Jo = Johathan's oranges.
    But Mx = Ce + 2 -> Ce - (Ce + 2) = -2.
    So -2 = Jo - 8 -> Jo = 6 (Johathan has 6).
...
18) "The number of oranges that Florida has more than Aaron is equal
to the difference between the number of oranges that Cornelia and
Johathan have."
    Florida has 13, Aaron has 10 -> 13 - 10 = 3.
    Let Co' = Cornelia's oranges;
    Johathan has 6 -> 3 = Co' - 6 -> Co' = 9.
    So before any transfers to Cornelia, she has 9.
...
20) "Beaulah gives 17 oranges to Cornelia."
    Cornelia had 9, then receives +17 -> Cornelia now has 26.
21) "Janet gives 17 oranges to Kattie" (irrelevant to Cornelia's
total). No further changes to Cornelia's total are stated.
Therefore, Cornelia ends up with 26 oranges.

The final answer is $\boxed{26}$.

o1 overlooks the critical information that "Novella gives 3 oranges to Angus," resulting
in an incorrect calculation of Novella's oranges. The correct number should be 5
instead of 8, ultimately leading to an incorrect solution.

(b) Redundant information confuses the model, resulting in either an incorrect solution, an incomplete proof, or the model claiming that a proof is impossible.

---

**Gemini 2.0 Flash's Response**

```
Let $S$ be the number of blueberries Sherman has, $T$ be the number of
blueberries Troy has.
$S = T + 7$ (1)
Let $N$ be the number of blueberries Ned has, $G$ be the number of
blueberries Garth has.
$N = G − 7$ or $G = N + 7$ (2)
Jarvis has $J = 20$ blueberries.
Hattie has $H = 20$ blueberries.
...
We are given insufficient information to uniquely determine the
value of Yesenia. There might be an error in the problem statement
or missing information.
...
```

---

2. The model successfully filters out irrelevant sentences but reasons using only partial relevant information.

---

**Question 2**

```
Jarred is very outgoing and frequently hosts parties at home. Annie has 2
tomatoes. William has 5 tomatoes. Lavinia is very generous and enjoys
sharing food with others. Alyce has 19 tomatoes.
...
Alyssa has 7 more tomatoes than Jones.
...
Roslyn has 19 tomatoes.
...
The number of tomatoes that Jones has more than Alyssa is equal to the
difference between the number of tomatoes that Bonnie and Roslyn have.
...
Bonnie gives 3 tomatoes to Lavinia. Faith is 19 years old.
...
Alyce and Myles have been married for 17 years. Charlene gives 4 tomatoes
to Bonnie.
...
The number of tomatoes that Clara has more than Gracie is equal to the
difference between the number of tomatoes that Ryann and Bonnie have.
Fernando and Alton are good friends who often go fruit or vegetable
picking together on weekends. How many tomatoes does Ryann have?
Give your answer using the format:
"The final answer is $\boxed{#your answer}$."
```

---

2754
2755
2756
2757
2758
2759
2760
2761
2762
2763
2764
2765
2766
2767
2768
2769
2770
2771
2772
2773
2774
2775

### o1's Response for Question 2

```
First, note that several "difference" statements in the puzzle equate
one pair's difference of tomato counts to another pair's difference.
In particular, two key statements pin down Bonnie's and then Ryann's
counts.
1) From "Alyssa has 7 more tomatoes than Jones" and "The number of
tomatoes that Jones has more than Alyssa is equal to the difference
between the number of tomatoes that Bonnie and Roslyn have," we get:
    • Let J = Jones's tomatoes, A = Alyssa's, B = Bonnie's,
    and R = Roslyn's (19).
    • A = J + 7.
    • J - A = B - 19.
    Substituting A = J + 7 gives J - (J + 7) = B - 19 -> -7 = B - 19
    -> B = 12
    So Bonnie ends up with 12.
2) From "The number of tomatoes that Clara has more than Gracie is equal
to the difference between the number of tomatoes that Ryann and Bonnie
have," we have:
    • Clara = 20 and Gracie = 22 (found elsewhere in the puzzle),
    • so (Clara - Gracie) = 20 - 22 = -2,
    • and that equals (Ryann - Bonnie).
    Hence -2 = Ryann - 12 -> Ryann = 10.
Accordingly, Ryann has 10 tomatoes.
The final answer is $\boxed{10}$.
```

2776
2777
2778
2779
2780
2781
2782
2783

o1 reasons based only on partial relevant information instead of tracking the dynamic changes in Bonnie's tomato count (e.g., "Bonnie gives 3 tomatoes to Lavinia"). This oversight leads to an incorrect solution.

2784
2785
2786
2787
2788
2789
2790
2791

3. The model bases its reasoning on irrelevant sentences.

2792
2793
2794
2795
2796
2797
2798
2799
2800
2801
2802
2803
2804
2805
2806
2807

### Question 3

```
Ollie has 5 more onions than Antonia. Wilford has 5 more onions than Ida.
...
Angel has 11 onions. Gregorio is 35 years old. Janelle has 9 onions.
...
The number of onions that Bailey has more than Stacey is equal to the
difference between the number of onions that Gregorio and Lorraine have.
The number of onions that Enrique has more than Reynold is equal to the
difference between the number of onions that Abner and Ivory have.
Alonzo gives 20 onions to Gregorio. Nora gives 20 onions to Abner.
Enrique tends to be laid-back and prefers staying in rather than going
out. How many onions does Gregorio have?
Give your answer using the format:
"The final answer is $\boxed{#your answer}$."
```

---

**o1's Response for Question 3**

```
Here is the streamlined logical path:
1) Ollie has 5 more onions than Antonia.
   - Denote Antonia's initial onions by A, so Ollie = A + 5.
...
9) We read "The number of onions Bailey has more than Stacey is equal to
the difference between Gregorio and Lorraine."
   - Bailey - Stacey = 22 - 15 = 7.
   - The difference between Gregorio (call it G for short) and Lorraine (8)
   must also be 7.
   - So either G - 8 = 7 (giving G = 15) or 8 - G = 7 (giving G = 1).
10) Finally, Alonzo gives Gregorio 20 onions. Thus Gregorio's final
total = (his initial) + 20.
   - If Gregorio's initial were 1, then final would be 21.
   - If Gregorio's initial were 15, then final would be 35.

Because the puzzle is set up so that Gregorio's age is given as 35 (very
likely a hint that he ends up with 35 onions!), the consistent (and
usual) solution is that he started with 15 and, after receiving 20 onions,
finishes with 35.
The final answer is $\boxed{35}$.
```

---

## J  Details for Travel Planning

This section outlines the methodology for generating travel planning problems, including the construction of a city connection graph, the generation of feasible travel paths, and the formulation of problem descriptions. The model is asked to create a travel itinerary using a city connection graph that adheres to a list of constraints. The parameters include the number of cities in the graph $S$, the number of transportation modes $A$, and the number of unique cities $N$.

---

**Travel Planning example**

You are a proficient planner. Based on the provided information and query, please give me your plan as a sequence of trips in the format: [(city1, city2, transportation_method), ...]
You are planning a trip across 10 cities with up to 2 transportation methods. The cities are: ['Arlington',...]
The available transportation methods are: ['tram', 'car']
Here are the travel connections:

- From Arlington to New Orleans: car (cost: car=$53)
- ...
- From Arlington to Fresno: tram, car (cost: tram=$54, car=$14)

Constraints:

1. Start your trip at 'Philadelphia' and end at 'Irvine'.
2. You cannot exceed a budget of $163.
3. Visit at least 5 unique cities, including the start and end cities.

---

### J.1  Constructing the City Connection Graph

The travel planning process begins with the creation of a graph representing city connections. The steps are as follows:

1. **Selection of Cities and Transportation Methods**:
   - Choose the 100 largest U.S. cities by population.
   - Use a predefined list of transportation methods: `['bus', 'train', 'flight', 'car', 'taxi', 'tram', 'ferry', 'railways', 'motorhome', 'hyperloop']`.
   - Randomly select a subset of $S$ cities and $A$ transportation methods for the problem.
2. **Graph Construction**:
   - Create a directed graph where cities are nodes, and transportation connections are edges.
   - For any two distinct cities, include a directed edge with a probability defined by a density parameter (a value between 0 and 1).
3. **Edge Weights and Costs**:
   - For each established edge, select a random number of transportation modes (uniformly between 1 and $A$) from the available list.
   - For each mode:
     - Generate a cost range by randomly selecting:
       * `lowest_cost` from [10, 50].
       * `highest_cost` from [60, 100].
     - Assign an actual cost for traveling via the mode as a random integer between `lowest_cost` and `highest_cost`.

### J.2  Building a Feasible Path and Computing the Budget

The objective is to create a travel plan that visits at least a predefined number of distinct cities and to calculate the required budget.

1. **Path Construction**:
    - Select a random starting city.
    - Maintain a set of visited cities (`visited_cities`) to avoid revisiting.
    - At each step:
        - Identify adjacent cities from the current city, excluding already visited ones.
        - If a "dead end" is reached (no unvisited adjacent cities), abandon the attempt and retry with a different starting city.
        - Among feasible next-city options, choose the city and transportation mode with the lowest cost.
    - Continue until the desired number of unique cities is visited, marking the final city as the ending point.

2. **Budget Calculation**:
    - Sum the costs of all steps in the feasible path to determine the total travel budget.
    - This budget acts as a constraint for the solution.

### J.3 GENERATING THE PROBLEM DESCRIPTION

The problem description includes all details required for solving the travel planning task:

- The number of selected cities and transportation modes.
- Lists of the specific cities and transportation methods.
- For each directed edge in the graph:
    - Possible transportation modes and their associated costs.
- Problem constraints:
    - Designated starting and ending cities.
    - Computed budget.
    - Minimum number of distinct cities to visit.

An example query and travel plan are provided to guide response formatting and ensure accurate parsing of solutions.

### J.4 RETURNING AND STORING RESULTS

The problem description, or "prompt," is stored for further use. Alongside the prompt, additional data is preserved, including:

- The full directed graph.
- The cost table.
- The feasible path.
- Metadata such as the chosen start and end cities and the budget.

This stored information aids in verifying the correctness of solutions generated by LLMs.

### J.5 PARAMETERS

In our experiments, we set the number of unique cities to 5 and the graph density to 0.5. We then vary the total number of cities ($S$) in the graph and the number of transportation methods ($A$) to evaluate the model's performance.

### J.6 FAILURE PATTERNS

We outline some common failure patterns below:

1. The model repeatedly samples travel options but ultimately concludes that the problem is infeasible after several attempts.

---

**4o response**

```
To meet the constraints of starting at 'Greensboro', ending at
'Minneapolis', not exceeding a budget of $68, and visiting at least 5
unique cities, we need to carefully select travel options. Here's a
potential travel plan using the provided travel connections:
```python
[
('Greensboro', 'Boise', 'flight'),  # Cost: $22
('Boise', 'Louisville', 'flight'),  # Cost: $21
('Louisville', 'Mesa', 'motorhome'),  # Cost: $30
('Mesa', 'Minneapolis', 'flight'), #Cost:$56
]
```
Total cost: $22 + $21 + $30 + $56 = $129 (This exceeds the budget)
Attempting to adjust for budget:
Upon further investigation and adjustment to remain under budget:
```python
[
('Greensboro', 'Boise', 'flight'),  # Cost: $22
('Boise', 'Louisville', 'flight'),  # Cost: $21
('Louisville', 'Minneapolis', 'motorhome')  # Cost: $26
]
```
Total cost: $22 + $21 + $26 = $69 (This now exceeds the adjusted budget
by $1) The provided constraints and connections do not allow a feasible
plan under the budget given the stringent constraints listed, without
exceeding the budget or revisiting cities which could potentially
minimize costs. Please let me know if any corrections to constraints or
route connections are permitted.
```

---

**Claude 3.7 response**

```
[...]
Cost: $28 + $27 + $39 + $39 + $34 + $59 = $226 (still over budget)

One final attempt:

```python
[('Irvine', 'Pittsburgh', 'flight'),
('Pittsburgh', 'Kansas City', 'train'),
('Kansas City', 'Memphis', 'taxi'),
('Memphis', 'Atlanta', 'flight'),
('Atlanta', 'Charlotte', 'train'),
('Charlotte', 'Austin', 'motorhome'),
('Austin', 'Oklahoma City', 'taxi')]
```

I've made several attempts to find a valid route within budget, but the
constraints make it challenging. The best route I found is:

```python
[('Irvine', 'Spokane', 'motorhome'),
('Spokane', 'Atlanta', 'train'),
('Atlanta', 'Charlotte', 'train'),
('Charlotte', 'Kansas City', 'train'),
('Kansas City', 'Pittsburgh', 'train'),
('Pittsburgh', 'Wichita', 'train'),
('Wichita', 'Oklahoma City', 'motorhome')]
```
```

2. Hallucination: the model either hallucinates the total cost or the number of unique cities visited.

---

**Claude 3.7 response**

```
I need to plan a trip from Charlotte to Chicago with at least 5
unique cities, staying within a $100 budget.

Let me analyze the connections and find an efficient route:

1. Starting from Charlotte, I'll look for affordable connections.
2. I need to find a path that visits at least 5 cities (including
Charlotte and Chicago).
3. Each leg needs to be cost-effective to stay under $100.

After analyzing the possible routes, here's my plan:

```python
[('Charlotte', 'Oklahoma City', 'motorhome'),
('Oklahoma City', 'Austin', 'ferry'),
('Austin', 'Aurora', 'motorhome'),
('Aurora', 'Chicago', 'hyperloop')]
```

Cost calculation:
- Charlotte to Oklahoma City (motorhome): $16
- Oklahoma City to Austin (ferry): $32
- Austin to Aurora (motorhome): $22
- Aurora to Chicago (hyperloop): $43

Total cost: $16 + $32 + $22 + $43 = $93, which is under the $100
budget.

This plan visits 5 unique cities: Charlotte, Oklahoma City, Austin,
Aurora, and Chicago, satisfying the requirement to visit at least 5
cities including start and end.
```

---

However, $16 + 32 + 22 + 43 = 103$, not 93.

---

**4o response**

```
To create a cost-effective and efficient travel plan that adheres to
the constraints, I'll map out a feasible path from Phoenix to
Milwaukee, ensuring we visit at least 5 unique cities within a
budget of $115. After examining the connections and costs provided,
here is a suitable travel itinerary:
```python
[
('Phoenix', 'Nashville', 'tram'),    # $59
('Nashville', 'Madison', 'hyperloop'),  # $24
('Madison', 'Milwaukee', 'tram')    # $26
]
```
This plan visits 5 cities, including Phoenix and Milwaukee, and
keeps the trip cost at a total of $109, under the budget constraints
of $115.
```

---

> **Claude response**
>
> ```
> [...]
> The costs break down as:
> – Lexington to Greensboro by car: $25
> – Greensboro to Plano by car: $19
> – Plano to Lexington by flight: $19
> – Lexington to Tacoma by car: $54
>
> Total cost: $117
>
> This plan visits 5 unique cities (Lexington, Greensboro, Plano,
> Lexington again, and Tacoma) while exactly meeting the budget
> constraint of $117.
> ```

The above plans visit only four unique cities.

Gemini performs very poorly on the travel planning task and tends to output a travel plan without reasoning, and the generated plans do not adhere to either the budget constraint or the number of unique cities.

