# OpenReview forum: "Frontier LLMs Still Struggle with Simple Reasoning Tasks"
_ICLR.cc/2026/Conference — Submitted to ICLR 2026_

### Official Review · Reviewer_BjBt · 2025-10-30

**Soundness:** 3
**Presentation:** 2
**Contribution:** 2
**Rating:** 2
**Confidence:** 4

**Summary:**

The paper looks at “simple but tedious” reasoning: a set of procedurally generated tasks (logic eval/negation, counting, proof-tree math, travel planning) plus UNPUZZLES and context-shift (CS) variants. The main message is that frontier/“thinking” LLMs still crumble when problems get longer or when surface form shifts, even when conceptual difficulty is fixed.

**Strengths:**

1. The datasets are simple, controllable, and easy to extend; I like that you can scale tediousness without changing the core concept.

2. UNPUZZLES + CS is a clever way to probe memorization and context corruption.

**Weaknesses:**

1. **Novelty/positioning.** Most insights are already known (long-context distraction, shortcutting, tokenization/copy issues, weak state tracking). The paper acknowledges related work but doesn’t push understanding deeper. As a result, the contribution is hard to pin down beyond a well-engineered bundle of tasks.


2. **Structure.** Section 3 (procedural tasks) and Section 4 (UNPUZZLES) read like two separate papers. If UNPUZZLES is mainly about memorization/educated guesses, say that explicitly and connect it to Section 3’s failure modes. Right now the bridge is thin.


3. **Results presentation.** The paper talks about trends “across LLMs,” but there are no per-setting cross-model aggregates (mean±std, or ranks). Readers have to scan tables model by model. Some table ordering is counter-intuitive (e.g., Logic Evaluation has (n=16) above (n=8)); one expects easier settings to appear first. Small thing, but it slows reading.


4. For each claim in §3.2 (Failure Analysis), the factors aren’t well controlled; for example, context length co-varies with tokenization effects. Consider padding prompts to a fixed length to isolate variables.


Useful engineering and a nice unpuzzle/CS idea, but the novelty is limited and several headline claims aren’t causally nailed down. With tighter controls, cross-model aggregates, and a cleaner §3↔§4 bridge, this could become a strong empirical paper.

**Questions:**

1. Line 363 claims that 20-character random names should hurt due to multi-token variables. However, the Logic-Negation results don’t consistently reflect this: at (d=12), 3 models favor random-20, 4 favor movies, 1 ties; at (d=8), random-20 wins 3 vs. 5 for movies; at (d=4), there are 3 ties, with random-20 winning 2 vs. 3 for movies. Could you clarify whether the degradation is statistically significant and, if so, how you attribute it specifically to multi-tokenization rather than other factors?

---

> ### Author Response · Authors · 2025-11-27
>
> Thanks for your constructive review – all the points you raised are valuable. Below we address your main criticisms. Please also see our "Joint response".
>
> 1. Novelty/positioning: While running specific experiments to target every failure mode we described is out of scope for a rebuttal, we did expand our analysis significantly by systematically categorizing the failures for every incorrect response. We feel this analysis provides a much more complete picture of the failures and points to many specific, actionable insights. For more details about our main contributions, please see our joint response.
>
> 2. Connection between procedural benchmarks and unpuzzles: done, see joint response.
>
> 3. As suggested, we tried to improve the presentation. Due to the lack of time we have not managed to provide meaningful cross-model aggregates, but we intend to do it (e.g., along the lines of the sketched error-rate analysis for Reviewer NJRb) for the final version (and perhaps towards the end of the discussion period).
>
> 4. While we agree, designing specific experiments to test specific failure causes seems difficult for closed-source models. Context length in particular is correlated with multiple other factors: tokenization, problem difficulty (reasoning steps), and the amount of irrelevant information. We have instead clustered the common “symptoms” of failure, such as omitting information, and making procedural calculation errors.
>
> Question 1: Tokenization: We agree that the adverse effect of random 20-character variable names might have been overstated. In the new version we used auto-evaluation to quantify the effects of tokenization, and now it only appears as a reason for failure in the counting problems, and even in the word-counting case it is an infrequent cause.

---

### Official Review · Reviewer_JRM9 · 2025-10-31

**Soundness:** 3
**Presentation:** 2
**Contribution:** 2
**Rating:** 4
**Confidence:** 5

**Summary:**

The paper tested state-of-the-art LLMs on a suite of procedurally generated tasks, including 1) counting, 2) first-order logic, 3) proof trees, and 4) travel planning. And a dataset UNPUZZLE that contains difficult puzzles, and their trivialized and context-shifted versions. With these tasks, the paper argues that reasoning LLMs exhibit similar drawbacks as non-reasoning models when solving these types of tasks.

**Strengths:**

The paper shows that state-of-the-art LLMs still struggle on simple reasoning tasks by constructing procedurally generated reasoning tasks and a human-annotated small puzzle dataset. It revealed several failure modes that are common among frontier LLMs.

**Weaknesses:**

The novelty of the paper is limited: the idea that frontier LLMs struggle with "simple"/"tedious" tasks has been studied in several works [1, 2, 3]. The advantage of this paper compared to previous works, e.g., the procedured generation, so that the benchmark is less prone to data contamination, does not provide new information about the fact that LLMs struggle with these simple tasks.

[1] Kazemnejad, Amirhossein, et al. "The impact of positional encoding on length generalization in transformers." Advances in Neural Information Processing Systems 36 (2023): 24892-24928.
[2] Yehudai, Gilad, et al. "When Can Transformers Count to n?." arXiv preprint arXiv:2407.15160 (2024).
[3] Dziri, Nouha, et al. "Faith and fate: Limits of transformers on compositionality." Advances in Neural Information Processing Systems 36 (2023): 70293-70332.

**Questions:**

1. The paper studied multiple closed-source models; on the proposed benchmark, these models have very different behaivours. It would be interesting and useful if more analysis were done on why these models differ from each other. E.g., for the models that generally perform better than average, is it more likely due to the model being trained with more data of this type of task, or due to the model having better reasoning generalizability as compared to other models?

2. What practical guidelines can we draw from the analysis of frontier model behaviours on the proposed benchmark?

3. The UNPUZZLE dataset suggests that the original well-known puzzles may already have appeared in the training data, thus leading to high performance. Similar investigations of data contamination were conducted in recent works, e.g., [1], where the method is simple: let the LLM complete a partial question, and see if it can successfully complete it and then answer. Can similar experiments be done on the original puzzles and see if the frontier LLMs already know the puzzle?

4. Why do context-shifted puzzles consistently have better performance compared to their unpuzzle version? If the reasoning behind this is: the unpuzzle version is more affected because it's similar to the original contaminated puzzle (lower score due to memorization of a different answer), while the context-shifted version is less affected. It seems to suggest that the LLMs do possess some degree of reasoning generalizability, as long as it's not affected by memorization? More discussion on the context-shifted experiments can be helpful for understanding this behaviour.

[1] Wu, Mingqi, et al. "Reasoning or memorization? unreliable results of reinforcement learning due to data contamination." arXiv preprint arXiv:2507.10532 (2025).

---

> ### Author Response · Authors · 2025-11-27
>
> Thank you for your valuable comments. Below we address your main criticisms. Please also see our "Joint response".
>
> - Novelty: see the joint response.
>
> 1. Analysis of failure causes: Given that training mixtures and training details are not available for most of the models, it is difficult to correlate these with evaluation results. While we cannot analyze the causes of failures, we have conducted a more thorough evaluation of the symptoms, by clustering failures into different types using an autorater.
>
> 2. Practical guidelines: we admit that the original paper did not propose many actionable insights except for what we could conclude from the context-shifted unpuzzles: if you rephrase the problem with new language, it is less likely that the model will be confused by reasoning patterns in its training data. However, we now include a comprehensive analysis for the simple reasoning problems, which identifies many specific, actionable changes. Please see the common paragraph for examples.
>
> 3. Puzzle continuation evaluation: We are confident that these puzzles are in the training data, as many appear on multiple websites including Wikipedia, and the models sometimes state that the puzzle is well-known in their response. The responses also mimic the ground-truth solutions on the web. However running the additional continuation evaluation is a good idea.
>
> 4. Yes, we believe that the models achieve better performance on the context-shifted unpuzzles because these are less affected by memorization. We will elaborate on this in the paper.

---

### Official Review · Reviewer_NJRb · 2025-10-31

**Soundness:** 1
**Presentation:** 1
**Contribution:** 3
**Rating:** 2
**Confidence:** 4

**Summary:**

This paper introduces a suite of procedurally generated simple reasoning tasks, including counting, first-order logic, proof trees, and travel planning, with changeable parameters and the UNPUZZLE dataset. Using these datasets, it performs an in-depth failure analysis across multiple SOTA LLMs, including thinking models.

**Strengths:**

In-depth error analysis.

Nice idea with the unpuzzle puzzles, especially the context-shifted unpuzzle, which allows for a more detailed failure analysis and attribution.

**Weaknesses:**

No human evaluation. You write “quite easy for humans,” but do not test this statement. You make this a central part of the paper, yet it remains untested.

The story should be cleaner. Sections 3 and 4 feel related, but currently they should be two papers. I suggest thinking more about how the paper is presented to tie these two together.

“One suggestion from our paper is that LLMs should be evaluated not only by the most difficult problem they can solve, but also by the simplest problem they struggle with.” This does not seem like a new idea, and is very much already part of the literature, as partially shown in the related work (see later comment).

No error bars. The experiments ought to be done multiple times to ensure the results are significant.

You should have tested newer models such as GPT5. The model selection is fine, but given the progress, it would be nice to see as recent models as possible.

“Similarly to other recent works, our results suggest that LLMs mimic training data rather than performing true reasoning, making it relatively easy to find out-of-distribution problems where the models fail.” What is then the novel contribution of this paper? While I like the idea of the paper, I find the evaluation and writing/presentation below the requirements for an ICLR paper.

While it is mentioned that "every model we tested performed better on the context-shifted unpuzzles than the original ones, indicating that failure was at least in part due to memorization of the original puzzle," this point is not made strongly enough and undermines the rest of the claims, and sometimes states the findings without context, which can be misleading. For example, in the abstract it is stated that:“... several systematic failure patterns related to memorizing the originals,” but a later statement does not connect strongly enough to this observation and instead sounds like a more general claim: "Our results highlight that out-of-distribution generalization is still problematic for frontier language models and the new generation of thinking models, even for simple reasoning tasks, and making tasks easier does not necessarily imply improved performance."


Minor:

The paper is missing some related works, such as https://arxiv.org/pdf/2407.06581 and https://arxiv.org/pdf/2504.12256.

You should remove Opus from the author list in Opus and Lawsen (2025). See the footnote in https://arxiv.org/abs/2506.09250v2.
Please include and reference examples of the failure cases listed at the end of Section 3.2.

Typos etc.:

Line 372 is missing a space.

Line 375 should be: See Appendix C for more details, dataset creation instructions, and some examples.

Line 464: .ressive results across a variety of

These are merely some highlights. Overall, the presentation of the paper is relatively poor.

**Questions:**

You have human annotators for unpuzzles and write “Each answer was assessed by a single annotator, or by consensus of all annotators if marked ambiguous.” For how many questions was this an issue?

You state that: "One of our goals was to design tasks that are easy (albeit tedious) for humans, but become unsolvable by frontier models when the difficulty parameters are large enough."
Could this instead be interpreted as a measure of robustness rather than mere difficulty?

You also argue that: "In general, if each computational step has a small probability of error, increasing the number of steps exponentially increases the probability of overall failure, even if the model follows the correct approach to calculating the solution."
Is the per-step failure rate somewhat constant, or is it clearly dependent on the number of previous steps? This again suggests that the model is not “worse” at longer tasks, just not robust.

Humans frequently make typos or small calculation mistakes in a tedious but easy math task. How do these results compare to human failure rates on tedious tasks qualitatively and quantitatively?

---

> ### Author Response · Authors · 2025-11-27
>
> Thank you for your valuable comments. Below we address your main criticisms. Please also see our "Joint response".
>
> - Comparison to humans: It is true that we have not tested humans on the tedious problems. However, our main statement (for the procedural tasks) is that the tasks are straightforward but tedious (in general). The unpuzzles are designed to be easy.
>
> - Connection between different parts of the paper: see joint response.
>
> - Main contributions: see joint response. We also replaced the cited sentence in the discussion with “This demonstrates that oftentimes LLMs mimic training data rather than performing true reasoning, making it relatively easy to find out-of-distribution problems where the models fail, and this problem is also present at the newest thinking models (while similar conclusions were hypothesized in other recent works, our result is the first to show this without actual access to the training data).”
>
> - Error bars are provided in the appendix for all reasoning tasks and all parameter settings.
>
> - Statements about memorization in the (un)puzzles problems: Since the context-shifted puzzles are designed to be less similar to the original puzzles that the normal puzzles, improved performance on them shows that memorization (similarity/overfitting to training data) affects model performance. In the abstract we changed “several systematic failure patterns” to typical failure patterns, which are demonstrated in Appendix E.
>
> - References: We have added the first suggested paper as, although in the visual domain, it deals with similar simple problems as we do. On the other hand, we felt that the other paper is somewhat out-of-scope of our paper. We have removed Claude Opus from the references.
>
> - Robustness vs difficulty: Robustness typically evaluates sensitivity to perturbations, whereas we evaluate sensitivity to the number of computation steps.
>
> - Error-rate vs. input length: Looking at the results, one can see that the failure rates increase with the length. For example, in the word-counting problem the (minimum) length of the problem is a reasonable approximation of the problem complexity, and hence one could expect that the probability of correctness on the longer problems is about the cube of the correctness probability of the shorter problems, but the performance drop is larger than this (Figure 3). Similarly, for the logic problems, the amount of computation changes exponentially with the depth, but the decrease in accuracy is faster than the resulting exponential decay (Figure 4).
>
> - Annotator disagreement: There was very little disagreement between annotators on the unpuzzle task, though we don't have exact numbers. We initially had more granular context corruption (CC) categories which led to some disagreement. This was solved by merging some of the initial categories into "correct with CC", "incorrect with CC" (delirium), and "incorrect (other)".

---

### Official Review · Reviewer_w54N · 2025-10-31

**Soundness:** 3
**Presentation:** 3
**Contribution:** 3
**Rating:** 6
**Confidence:** 3

**Summary:**

This paper evaluates state-of-the-art LLMs, including recent thinking models (o1, o3, DeepSeek R1), on procedurally generated "easy" reasoning tasks and introduces the UNPUZZLES dataset of trivialized logic puzzles. The authors demonstrate that even thinking models fail when task parameters increase tediousness while preserving fundamental difficulty, and that models paradoxically perform worse on simplified versions of puzzles they can solve in their original form.

**Strengths:**

S1. The paper provides a timely and comprehensive evaluation of thinking models on simple reasoning tasks, filling a gap in the literature since most prior work focused on earlier model generations. The inclusion of o1, o3, DeepSeek R1, and Gemini thinking variants makes this a valuable reference for understanding current capabilities.

S2. The procedurally generated task suite with tunable parameters (document length, tree depth, number of cities) is well-designed and allows systematic study of how performance degrades with increased computational requirements while difficulty remains constant. This methodology is reproducible and extensible to future models.

S3. The UNPUZZLES dataset introduces a novel and counterintuitive phenomenon where models fail on trivialized versions of puzzles they solve correctly. The "reasoning delirium" failure mode, where models apply memorized complex solutions to simple problems, is an important finding that reveals limitations in how models generalize.

S4. The context-shifted unpuzzles provide clever experimental control by demonstrating that models can solve problems with identical logic but different surface form, isolating memorization as a key factor in unpuzzle failures rather than fundamental reasoning inability.

S5. The failure analysis systematically categorizes error patterns (statistical shortcuts, error accumulation, long context difficulties, poor OOD generalization) with concrete examples from model outputs, making the findings actionable for future research.

**Weaknesses:**

W1. Several tasks conflate multiple difficulty dimensions simultaneously, making it unclear which factors drive failures. For example, increasing tree depth in logic tasks increases both the number of reasoning steps and context length. More controlled ablations isolating individual factors would strengthen causal claims about failure modes.

W2. The UNPUZZLES dataset, while introducing an interesting phenomenon, is limited in size (97 puzzles) and requires manual construction and evaluation. This limits statistical power and reproducibility. The reliance on human annotation for correctness and "context corruption" introduces subjectivity, with inter-annotator agreement not reported.

W3. The paper acknowledges that thinking models show quantitative improvements even while maintaining qualitative failure trends, but does not deeply engage with what these partial improvements mean. If o1 achieves 60% accuracy where GPT-4o achieves 30%, this suggests meaningful progress even if imperfect, yet the framing emphasizes failure.

W4. The practical implications of these findings remain unclear. The tasks, while procedurally elegant, are somewhat artificial (counting words in paragraphs, logic formulas with random variable names). Whether failures on these tasks predict failures on real-world reasoning problems is not established.

**Questions:**

Q1. The concurrent work by Shojaee et al. (2025) is noted to have design flaws. Could you elaborate on how your experimental design avoids similar issues, particularly regarding the claim that their problems may be unsolvable or ignore token limits?

Q2. For UNPUZZLES, how sensitive are results to the specific method of trivialization? If you created multiple different simplified versions of the same puzzle, would models fail consistently across all simplifications?

Q3. Some failure modes (tokenization issues, specific formatting expectations) seem potentially addressable through engineering improvements. Can you estimate what fraction of failures stem from such fixable issues versus fundamental reasoning limitations?

Q4. The thinking models often show substantial improvement over non-thinking variants even if not achieving perfect performance. How should the community interpret this partial progress? Does it suggest the approach is fundamentally sound but needs scaling, or that different approaches are needed?

Q5. In the UNPUZZLES evaluation, models are assessed once per puzzle. Given the known variability in LLM outputs, would results differ substantially with multiple samples and majority voting?

---

> ### Author Response · Authors · 2025-11-27
>
> Thank you for your valuable comments. Below we address the weaknesses and questions listed in the paper. Please also see our "Joint response".
>
> **W1 (conflated difficulties):**
> We have added a more granular auto-evaluation to isolate different sources of error. For the case mentioned by the reviewer (context length and reasoning steps both increase with tree depth), the autoevaluation includes “omission” and "procedural error” categories, corresponding to ignoring some information (potentially due to long context) and making a reasoning step error, respectively.
>
> **W2 (unpuzzles small, manual, subjective):**
> We take the reviewer’s point that the set of puzzles and unpuzzles is small. Unfortunately there aren’t too many logical puzzles that are famous enough to be memorized by models to such an extent. For reproducibility purposes, we plan to release both the puzzles and unpuzzles. While some questions ask for strategies, the majority of them (the subset corresponding to context-shifting experiments) have numerical or categorical answers which can be checked automatically without human annotators.
>
> **W3 (acknowledging improvements of thinking models):**
> As the reviewer states, we do acknowledge the superior performance of thinking models on our benchmarks. It is difficult to engage with what this means without speculation. Broadly we think this can be explained by the fact that our problems require the correct execution of a sequence of simple steps, which seems aligned with the way “thinking” models are trained. Our new granular evaluation shows that thinking models take fewer shortcuts when solving problems, and commit fewer omission errors.
>
> **W4 (unclear practical implications):**
> LLMs have widely been observed to fail on very simple “real world” tasks, and our work attempts to systematize these failures. We aim to address this shortcoming with our systematic categorization of model errors, as described in the common paragraph. Finding frequent and  specific failures (often specific to a particular model) is very actionable, as these failures can be fixed in pre- or post-training once identified, leading to improved models.
>
> **Q1 (unsolvable problems, token limits):**
> All of our problems are solvable by construction, and we have not run into token limits when querying any models. (However, we have come across models stating that the problem is too long and that they will provide an educated guess.)
>
> **Q2 (different trivializations):**
> For most puzzles, it seems difficult to come up with many meaningfully different trivializations, especially since we did this work manually. Many trivializations involve removing a constraint that makes the puzzle difficult, such as increasing boat capacity in ‘river crossing’. But the failures are consistent e.g. for different sufficiently large boat capacities.
>
> **Q3 (fraction of engineering vs reasoning limitations):**
> We believe that most of the failures arise from reasoning step errors, rather than easily-fixable limitations such as tokenization and formatting.
>
> **Q4. (thinking models):** We believe that thinking models perform better due to breaking up problems into simple steps, which is aligned with the way our problems are set up. However, they still have a non-zero probability of making an error in each step. Their performance is potentially improvable using techniques like stepwise verifiers and self-revision. In our new error analysis, we notice that thinking models make fewer errors due to ‘shortcuts’ / educated guesses than other models, and omit less information.
>
> **Q5 (unpuzzles majority vote performance):** We will add an evaluation of pass@k; we suspect it will not result in a meaningful improvement.

---

### Author Response · Authors · 2025-11-27
**Joint response**

**Analyzing results and adding more SOTA models:**
We agree with the reviewers that the failure mode analysis for the simple reasoning tasks was anecdotal and not rigorous. To remedy this, we took advantage of new strong base models (namely Gemini 3.0 Pro) to help us identify errors and specific mistakes in the incorrect answers we collected from evaluating our dataset. We also added the recent models Gemini 3.0 Pro and GPT-5.1 to the procedural evaluations (the latter replacing GPT 4o, with which we identified some errors in our computations, although it seems that our queries to GPT-5.1 were often rerouted to less capable models).

The manuscript has been edited and uploaded to include these new results; we summarize them below for convenience.
While it is difficult to pin down the causes of the model failures, we have performed an LLM-assisted analysis of the failure symptoms in the answers and reasoning traces, identifying eight broad classes of errors: procedural, omission, parsing, copying, tokenization, hallucination, shortcut/heuristic, and abandonment.

Broadly, we found that all models make procedural errors, as well as omission errors. They also hallucinate quite a lot when the problem involves a composition of reasoning and natural language (ProofTree and Travel Planning) while hallucination is less of a problem in the clean logic and counting tasks. Importantly, we observed that thinking models are less prone to shortcuts, which demonstrates the real strength of producing reasoning traces. We also found model-specific behaviors that suggest these models had been trained to possess somewhat different skills.  We hope that identifying specific failures leads to actionable, specific strategies for their correction.

These results are included in Section 4 of the revised manuscript (uploaded). We believe that this analysis addresses the most important concerns, and thank the reviewers for their constructive feedback.

**Connecting the two parts of the paper** (procedural benchmarks and unpuzzles)

We have also added a paragraph connecting the procedural evaluation and unpuzzles sections of the paper - see the last paragraph of Section 4 in the updated version: One of the reasons behind the failures observed in the procedural tasks is identified as memorization or overfitting, and the Unpuzzles dataset is designed to examine exactly this issue.

**Main contributions**
1. To our knowledge, our paper is the first to demonstrate that even thinking models fail on many types of easy but tedious tasks and trivialized puzzles (appearing sporadically in the literature and on the internet). We also provide a more systematic and broad study in this area than what is available in the literature.
2. With the unpuzzles we perform an in-depth and novel analysis which shows the effects of memorization in LLM learning. This analysis is applicable even without having access to the training data, which allows us to analyze (closed) SOTA models.
3. We provide (in the revised version) an analysis of the error types in solving the problems and identify specific advantages of thinking models (see above). We are not aware of the existence of similar results. We also believe that the automatic analysis procedure we applied is of independent interest (and demonstrates the capabilities of the newest models).
4. Our datasets can be used as benchmarks for future LLM development.

---

### Meta-Review · Area_Chair_4vm5 · 2026-01-06

**Summary:**

1) Reviewers noted that although the paper includes many experiments, the evidence does not fully support several strong claims. In particular, causal explanations of failure modes, distinctions between memorization and reasoning, and conclusions about the behavior of “thinking” models are not clearly demonstrated.

2) Multiple reviewers pointed out weaknesses in the experimental design. These include the lack of human baselines despite claims that tasks are easy for humans, limited statistical validation (such as repeated runs or variance reporting), and task designs where multiple difficulty factors are mixed, making failure causes hard to isolate.

3) Reviewers questioned whether the results go beyond existing work on LLM failures in simple or long-horizon reasoning tasks. They also raised concerns about clarity and structure, noting that the procedural benchmarks and UNPUZZLES sections are not well integrated.

**Reviewer Concerns:**

1) The rebuttal addresses this in part by adding newer models, expanding the error taxonomy, and clarifying the intended link between procedural tasks and UNPUZZLES. However, the analysis remains largely descriptive, and key conclusions still rely on interpretation rather than controlled evidence. This concern remains partially unresolved.

2) While the rebuttal adds some clarifications and supplementary analyses, core issues remain. These include the absence of human evaluation, limited statistical validation in the main text, and a lack of well-controlled ablations. As a result, confidence in the empirical conclusions remains limited.

3) The rebuttal improves the explanation of the authors’ intent and slightly refines the framing. However, concerns about incremental novelty and structural clarity remain. The contribution is still difficult to distinguish from prior work, and the integration of the paper’s components falls short of ICLR expectations. With one reviewer saying "The story should be cleaner. Sections 3 and 4 feel related, but currently they should be two papers." and another one "Section 3 (procedural tasks) and Section 4 (UNPUZZLES) read like two separate papers."

Furthermore, as of now the amended manuscript is longer than 10 pages, violating the 9+1 page rule, which might also be a reason for a desk rejection.

**Reviewer Scores:**

I am not sure as none of the reviewers engaged with the rebuttal.

---

### Decision · Program_Chairs · 2026-01-26

Reject